# Tight junction protein occludin regulates progenitor Self-Renewal and survival in developing cortex

Raphael M Bendriem[1,2], Shawn Singh[1], Alice Abdel Aleem[3], David A Antonetti[4], M Elizabeth Ross[1,2]*

[1]Center for Neurogenetics, Feil Family Brain and Mind Research Institute, Weill Cornell Medicine, New York, United States; [2]Graduate School of Medical Sciences, Weill Cornell Medicine, New York, United States; [3]Weill Cornell Medicine Qatar, Doha, Qatar; [4]Kellogg Eye Center, Ophthalmology and Visual Sciences, University of Michigan Medical School, Ann Arbor, United States

**Abstract** *Occludin* (*OCLN*) mutations cause human microcephaly and cortical malformation. A tight junction component thought absent in neuroepithelium after neural tube closure, OCLN isoform-specific expression extends into corticogenesis. Full-length and truncated isoforms localize to neuroprogenitor centrosomes, but full-length OCLN transiently localizes to plasma membranes while only truncated OCLN continues at centrosomes throughout neurogenesis. Mimicking human mutations, full-length OCLN depletion in mouse and in human CRISPR/Cas9-edited organoids produce early neuronal differentiation, reduced progenitor self-renewal and increased apoptosis. Human neural progenitors were more severely affected, especially outer radial glial cells, which mouse embryonic cortex lacks. Rodent and human mutant progenitors displayed reduced proliferation and prolonged M-phase. OCLN interacted with mitotic spindle regulators, NuMA and RAN, while full-length OCLN loss impaired spindle pole morphology, astral and mitotic microtubule integrity. Thus, early corticogenesis requires full-length OCLN to regulate centrosome organization and dynamics, revealing a novel role for this tight junction protein in early brain development.

**\*For correspondence:**
mer2005@med.cornell.edu

**Competing interests:** The authors declare that no competing interests exist.

## Introduction

Mutations in the occludin (*OCLN)* gene cause a recessively inherited severe human disorder of microcephaly and band-like calcifications with polymicrogyria (BLC-PMG) characterized by loss of cortical convolutions, shallow or absent sulci, and multiple small gyri giving the cortex surface a roughened irregular appearance (*Abdel-Hamid et al., 2017*; *O'Driscoll et al., 2010*; *Jenkinson et al., 2018*; *Aggarwal et al., 2016*; *Elsaid et al., 2014*). The integral tight junction (TJ) protein, occludin, is recognized as part of epithelial and endothelial junctional complexes (*Furuse et al., 1996*; *Balda, 1996*; *Van Itallie and Anderson, 1997*; *McCarthy et al., 1996*) and while not required for tight junction assembly (*Saitou et al., 2000*; *Schulzke et al., 2005*), recent data indicates occludin regulates barrier properties (*Bolinger et al., 2016*; *Raleigh et al., 2011*). In the embryonic cerebral cortex, OCLN is localized both at TJs between epithelial cells and at the apical surface of the ventricular zone (VZ) in the chick (*Aaku-Saraste et al., 1996*). However, OCLN function in cortical development is virtually unexamined, as its VZ expression was thought to be turned off around the neuroepithelial to radial glial cell (NE-to-RGC) transition (embryonic day 11–12 (E11-E12) in the mouse), at the onset of neurogenesis (*Aaku-Saraste et al., 1996*; *Sahara and O'Leary, 2009*). A mouse model presumed to be an *Ocln*-null displayed postnatal growth retardation, brain calcification, and chronic inflammation. However, vascular endothelial TJs were still present (*Saitou et al., 2000*), suggesting that *Ocln* is important for regulation of TJs but not for junction

formation. Cortical phenotypes of this mouse model were not explored beyond scattered calcification in the brain.

Interestingly, mutations in other TJ protein-encoding genes such as *JAM3* produce brain hemorrhage (*Mochida et al., 2010*) rather than the congenital microcephaly and PMG associated with the *OCLN* mutation phenotype in humans, suggesting that OCLN may have developmental functions unanticipated for a TJ protein. Microcephaly, defined as head circumference of −2 standard deviations below the mean or smaller, can occur when expansion of the neural progenitor pool and subsequent generation of neurons is restricted (*Kaindl et al., 2010*; *Manzini and Walsh, 2011*; *Mahmood et al., 2011*; *Thornton and Woods, 2009*). Among the extensive list of genes now causally linked to microcephaly, many have in common a role in regulating centrosome dynamics and mitotic spindle stabilization of progenitors in the ventricular neuroepithelium (*Gilmore and Walsh, 2013*; *Faheem et al., 2015*; *Jayaraman et al., 2018*). In this study, we use human and mouse models of corticogenesis to explore the role of OCLN in the developing cortex, specifically to investigate its potential interaction with the centrosome and elucidate mechanisms through which its loss-of-function produces microcephaly.

We use mouse and human models to show that OCLN functions in cortical development, playing a previously unappreciated role in neural progenitor proliferation through promoting centrosomal and mitotic spindle integrity. Specific loss of the full-length OCLN isoform results in altered spindle and astral microtubules, prolonged M-phase, premature cell cycle exit and early neuronal differentiation. These defects are consistent with observed microcephaly and PMG associated with human *OCLN* mutations.

## Results

### OCLN localizes to interphase and mitotic centrosomes in embryonic mouse cortex

It is widely held that OCLN functions in tight junctions and its expression in the embryonic cortex is limited to neuroepithelial (NE) junctions (*Götz and Huttner, 2005*) prior to the NE-to-RGC transition at the onset of neurogenesis, at which point OCLN expression is believed to be turned off (*Aaku-Saraste et al., 1996*; *Sahara and O'Leary, 2009*; *Götz and Huttner, 2005*). This limited OCLN expression at tight junctions would be counterintuitive to the severe human microcephaly associated with *OCLN* mutation, since primary microcephaly is predominantly caused by defects in progenitor proliferation in cortex during the neurogenetic epoch, E11-E18 in the mouse [reviewed in 24]. We therefore sought to determine whether alternative subcellular expression of mouse OCLN (mOCLN) existed in the VZ before and after the NE-to-RGC transition. At E10.5, mOCLN is localized to the neuroepithelial plasma membranes and at the centrosome during interphase (*Figure 1A*) and mitosis (*Figure 1B*). By E14.5, mOCLN was absent at the cell membrane, in accordance with previous studies in which TJs are replaced by adherens junctions as neurogenesis begins (*Aaku-Saraste et al., 1996*). Surprisingly, mOCLN expression at both interphase and mitotic centrosomes persisted throughout and beyond the NE-to-RGC transition (*Figure 1A,B*). Several mouse *Ocln* transcripts are annotated in the Ensembl database: (1) full-length transcripts with varying 3' and 5'UTR lengths, all encoding wild-type OCLN (mOCLN-FL) with four transmembrane domains, two extracellular loops with intracellular N- and C-termini, and (2) one truncated transcript (ENSMUST00000159459) lacking exons 2 and 3, resulting in the loss of the N-terminus and three of the four transmembrane domains (mOCLN-ΔN; UniProt E0CZ73) (*Figure 1C*). Centrosomal localization of both isoforms was further supported by centrifugation-based subcellular fractionation of whole brain E12.5 lysates, which showed mOCLN-FL residing in both membrane and soluble nuclear fractions, with the latter fraction also containing centrosomal proteins γ-tubulin and phosphorylated-β-catenin (*Figure 1D*). In contrast, truncated mOCLN-ΔN could only be found in the soluble fraction. Together, these results indicate both OCLN isoforms localize to the centrosome while only mOCLN-FL localizes to the plasma membrane. Furthermore, although mOCLN-FL expression is downregulated after the NE-to-RGC transition, the truncated mOCLN-ΔN isoform is continually expressed during neurogenesis.

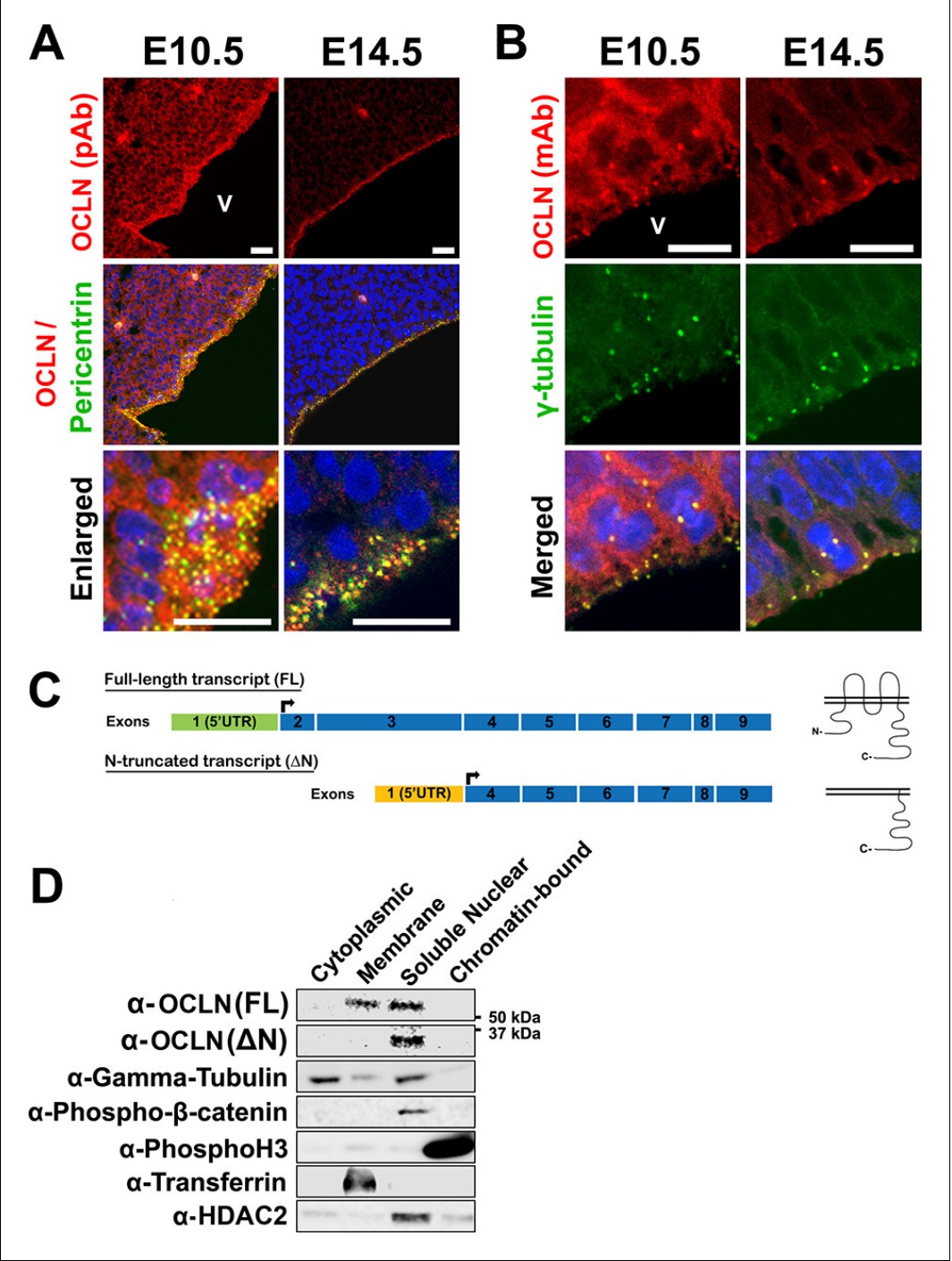

**Figure 1.** Differential localization of OCLN isoforms in embryonic mouse cortex. (**A,B**) Confocal microscope images of coronal sections from E10.5 and E14.5 mouse cortices stained with polyclonal (pAb) (**A**) and monoclonal (mAb) (**B**) OCLN antibodies and co-stained with centrosomal markers pericentrin or γ-tubulin. V, ventricle. Scale bar, 25 μm in A (including enlarged images) and 10 μm in B. (**C**) Schematic of mouse *Ocln* transcripts and their corresponding presumed protein structure. (**D**) Subcellular localization of occludin by immunoblot analysis of subcellular fractions (cytoplasmic, membrane, soluble nuclear, and chromatin-bound). Centrosomal markers γ-tubulin and phospho-Beta-catenin colocalize with occludin in the soluble nuclear fraction. Fraction-specific controls used are anti-HDAC2 (soluble nuclear), anti-transferrin (membrane), and anti-Phospho-histone3 (chromatin-bound).

## *Ocln* mutant mice exhibit microcephaly

An *Ocln* loss-of-function mouse line was previously generated by excising exon three and was characterized as a homozygous null model (*Saitou et al., 2000*). However, novel mouse *Ocln* splice forms have been recently annotated in the mouse genome. Using unique 5'UTR primers to distinguish between *Ocln-FL* and -ΔN transcripts, we noted a shorter FL transcript in the mutant mouse, whose PCR amplicon sequence revealed a frame shift predicting translation of a nonsense peptide with premature stop codon (*Figure 2A,B*). Moreover, we further noted in both control and homozygous mutant mice the presence of the shorter ΔN transcript lacking exons 2 and 3 which would not have been targeted for deletion in the mutant mouse. This transcript is translated to mOCLN-ΔN, a 32 kDa protein detectable by western blot using an antibody raised against the C-terminus of OCLN (*Figure 2A,B*). The homozygous mutants were hence named $Ocln^{\Delta N/\Delta N}$ or ΔN/ΔN in mouse data presented. To validate these findings immunohistochemically in the developing cortex, we probed for OCLN using an N-terminus specific antibody that would solely recognize mOCLN-FL and not protein from the mutant mouse lacking the OCLN N-terminus. Unlike the $Ocln^{\Delta N/\Delta N}$ mutant VZ progenitors, mOCLN-FL was detected in wildtype, $Ocln^{+/+}$ mouse brain at E10.5, localized to both plasma membrane and centrosome, and was absent at the plasma membrane by E14.5 (*Figure 2C*). Faint, diffuse OCLN labeling was observed in some but not all $Ocln^{\Delta N/\Delta N}$ VZ nuclei in the developing cortex. Immunohistochemical labeling of mOCLN-FL was substantially reduced in VZ progenitor plasma membrane and centrosome at E12.5 (*Figure 2C*). However, mOCLN-FL protein expression on Western blot in E12.5 *Ocln +/+* total brain lysate remained strong, presumably due to the presence of tight junctions in the brain vasculature, choroid plexus and other epithelial and endothelial elements (*Luissint et al., 2012*; *Kaur et al., 2016*) (*Figure 2—figure supplement 1*). In contrast, both $Ocln^{+/+}$ and $Ocln^{\Delta N/\Delta N}$ mice exhibited centrosomal staining with OCLN C-terminus antibody (detects both mOCLN-FL and mOCLN-ΔN) at E10.5 and E14.5 in the developing cortex (*Figure 2D*). Thus, the mutant mice still express the shorter isoform mOCLN-ΔN but not mOCLN-FL. These data are in agreement with subcellular fractionation results showing that mOCLN-ΔN only localizes at the centrosome and not at the plasma membrane (*Figure 1D*). Although the original characterization of the *Ocln* mice showed significant postnatal growth retardation, cortex size was not specifically assessed (*Saitou et al., 2000*). As early as postnatal day 7 (P7) and persisting through P20, we observed a significant 8–12% reduction in cortical surface area in $Ocln^{\Delta N/\Delta N}$ brains (*Figure 2E,F*) as well as a reduction in cortex thickness (*Figure 2G,H*).

## Prolonged mitosis and apoptosis in *Ocln*-mutant mice

Two major recurring mechanisms have been identified that underlie microcephaly in mutant mouse models. The first is M-phase prolongation, which has been shown to influence radial glial cell (RGC) division outcomes, leading to more apoptotic events and early cell cycle exit, generating neurons at the expense of the progenitor pool (*Pilaz et al., 2016*; *Lizarraga et al., 2010*; *Marthiens et al., 2013*; *Chen et al., 2014*). Thus, we analyzed the mitotic index compared to the proliferation index of the *Ocln*-mutant, which serves as a rough indication of the length of M-phase (*Smith and Dendy, 1962*). We observed a significant increase in mitotic index, defined as the number of PH3+ M-phase cells per total number of cells, in E10.5 and E12.5, but not E14.5 *Ocln*-mutant cortices compared to wild-type (*Figure 3A,B*), consistent with a prolongation of M-phase in E10.5 and E12.5 mutants. Furthermore, similar to previous reports of prolonged M-phase (*Pilaz et al., 2016*), we observed a higher percentage of cells in $Ocln^{\Delta N/\Delta N}$ E12.5 cortices in prometaphase and metaphase (PM+M) compared to wild-type (*Figure 3C,D*). The increased mitotic index in the *Ocln*-mutant was not due to greater overall proliferation in the cortex since we observed significantly reduced numbers of Ki67-labeled, cycling cells in the E12.5 $Ocln^{\Delta N/\Delta N}$ mutant mice (*Figure 3E,F*), indicating that the increased mitotic index reflects a greater proportion of proliferative cells in M-phase. Taken together, these results point to a critical period of mOCLN-FL function, from the earliest cortical event of neural tube formation lasting until E12.5, as mOCLN-FL is downregulated, asymmetric divisions and the rate of neurogenesis increases (*Aaku-Saraste et al., 1996*).

A second recurring mechanism in mouse models of microcephaly is premature detachment of RGCs from the ventricular surface as a result of centriole duplication impairments and subsequent loss of centrosome and cilia (*Insolera et al., 2014*; *Jayaraman et al., 2016*). The attachment of RGCs to the ventricular surface was preserved since immunoreactivity of apical markers atypical PKC

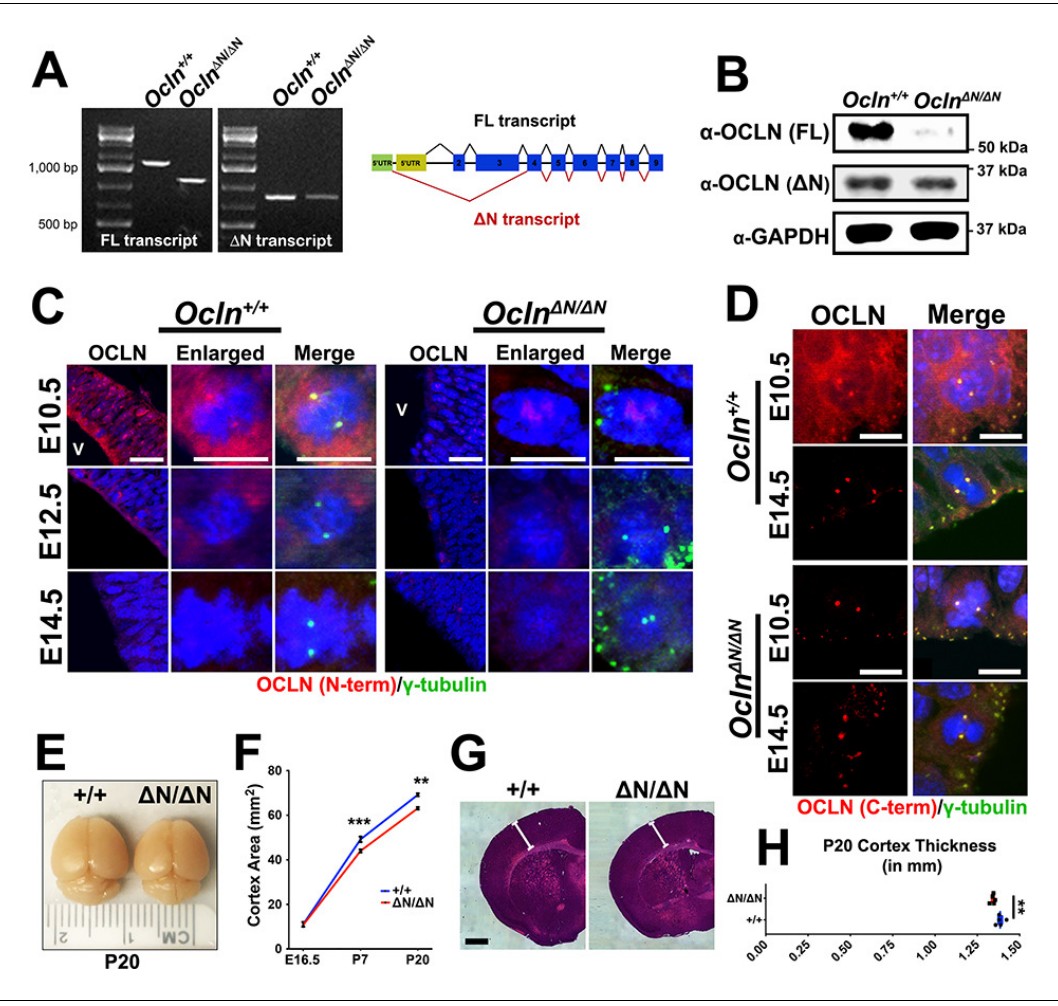

**Figure 2.** The *Ocln*^ΔN/ΔN mouse, a hypomorphic mutant line, exhibits microcephaly by P7. (**A**) Qualitative RT-PCR of *Ocln* expression in control and mutant E12.5 mouse probing for full-length (FL) and truncated (ΔN) transcripts. Representative image is shown from a sample of three biological replicates (primers in *Supplementary file 1*). (**B**) Western blot analysis of E12.5 mouse total brain lysate from *Ocln*^+/+ and *Ocln*^ΔN/ΔN mice. Representative image from three biological replicates. (**C,D**) Confocal images of coronal sections from E10.5-E14.5 control and mutant mouse cortices stained with N-terminus (**C**) and C-terminus-specific (**D**) occludin antibody and co-stained with centrosomal marker γ-tubulin. V, ventricle. Scale bar, 25 um in **C**) and 10 um in D. (**E**) Dorsal view of P20 control and mutant brains. (**F**) Quantification of cortical area of E16.5, P7, and P20 mouse brains (n = 4 brains for E16.5 control, n = 6 for E16.5 mutant, n = 8 for control and mutant P7, n = 3 for control and mutant P20). Data points represent mean ± SEM. Two-way ANOVA with Sidak's multiple comparison test; in each age group, +/+ was compared to ΔN/ΔN, **p<0.01,***p<0.001. (**G**) P20 brain section of wild-type and mutant mice cortices stained with H and E (Hematoxylin and Eosin). Scale bar, 1 mm (**H**) Quantification of P20 cortical thickness (n = 5 for each genotype). Data points represent mean ± SEM. Student's t-test, **p<0.01. Detailed tabulation of means, SEMs, sample sizes, and exact p-values can be found in *Figure 2—source data 1*.
The online version of this article includes the following source data and figure supplement(s) for figure 2:

**Source data 1.** Mean, SEM, sample size (n), and exact p-values for *Figure 2* quantifications.
**Figure supplement 1.** mOCLN-FL co-localizes with vascular endothelial marker, CD31, in *Ocln*^+/+ but not *Ocln*^ΔN/ΔN embryonic cortex.

(aPKC), mouse partitioning defective 3 (mPARD3), or β-catenin at the apical surface was indistinguishable between control and mutant cortices at E10.5 or E12.5 (*Figure 3—figure supplement 1A*). Furthermore, we did not see PAX6+ RGC displacement from the germinal zone at E12.5–16.5 (*Figure 4E,F*) nor did we observe more PH3-labeled M-phase cells in an abventricular position (*Figure 3A*), suggesting that RGCs are not prematurely delaminating from the ventricular surface.

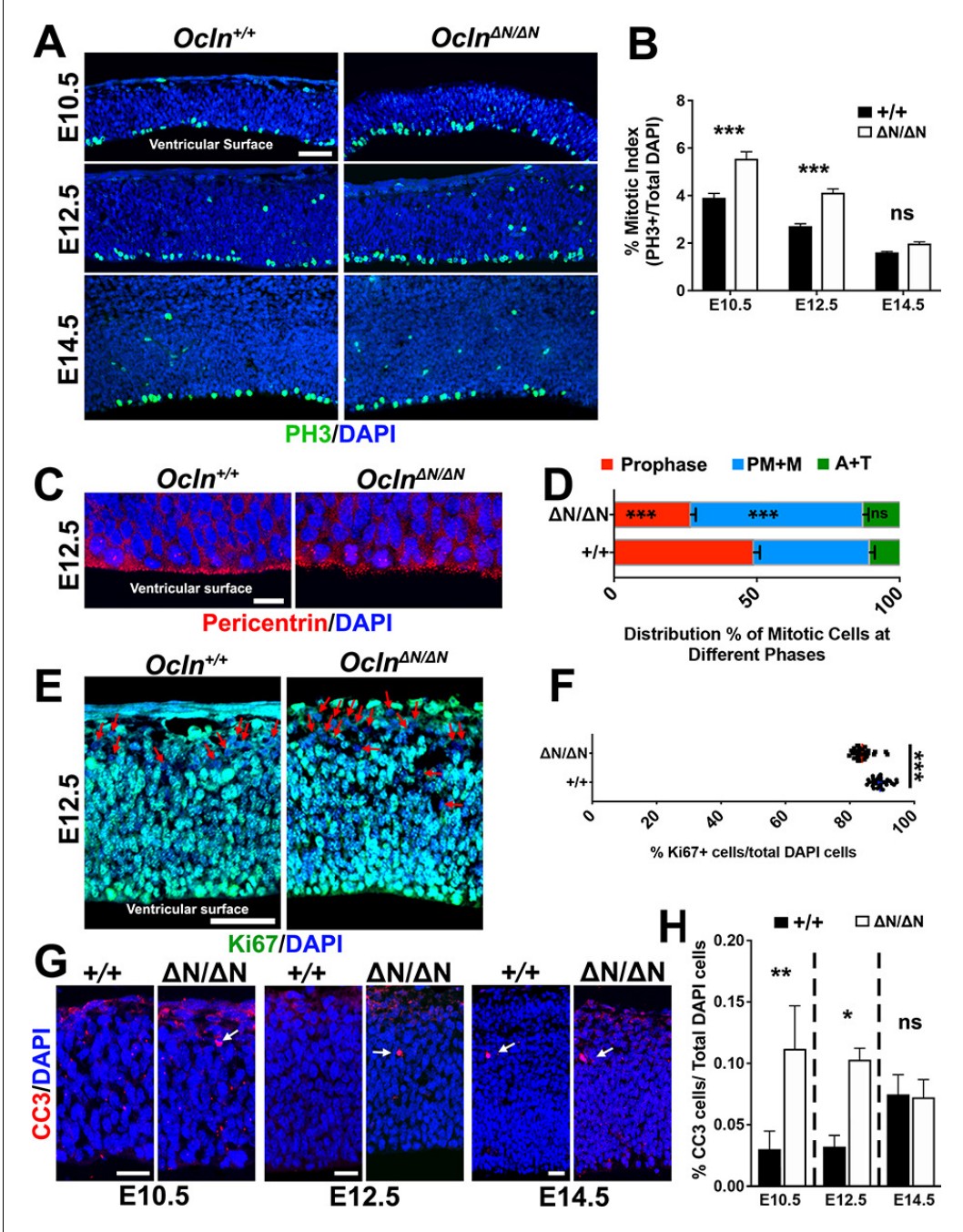

**Figure 3.** Progenitor proliferation abnormalities and apoptosis in $Ocln^{\Delta N/\Delta N}$ embryonic cortex. (**A**) Confocal microscope images of coronal section from E10.5-E14.5 control and mutant mouse cortices stained with M-phase marker phospho-histone3 (PH3; green) and counterstained with DAPI (blue). Scale bar, 50 μm (**B**) Mitotic index quantification, defined as the percentage of DAPI cells labeled with PH3, in $Ocln^{+/+}$ and $Ocln^{\Delta N/\Delta N}$ mice. All data points represent mean ± SEM, n = 30–39 brain sections for each genotype, from three independent experiments. Two-way ANOVA with Sidak's multiple comparison test; in each age group, +/+ compared to ΔN/ΔN, ***p<0.001, ns, not significant. (**C**) Confocal microscope images of coronal section from E12.5 $Ocln^{+/+}$ and $Ocln^{\Delta N/\Delta N}$ mouse cortices stained with centrosomal marker pericentrin (red) and counterstained with DAPI (blue). Phases of mitosis are identified by their unique DAPI and centrosomal staining patterns. Scale bar, 10 um. (**D**) Distribution percentages of M-phase stages in E12.5 control and mutant mice cortices at the ventricular surface. All data points represent mean ± SEM, n = 28 $Ocln^{+/+}$ sections and n = 29 $Ocln^{\Delta N/\Delta N}$ sections, compiled from three independent experiments. One-way ANOVA with Sidak's multiple comparison test comparing control and mutant data sets in

*Figure 3 continued on next page*

*Figure 3 continued*

each stage of mitosis, ***p<0.001, (E) Confocal images of E12.5 control and mutant mouse cortices in coronal sections stained with proliferation marker Ki67 (green) and counterstained with DAPI (blue). Red arrows denote Ki67-negative cells. Scale bar, 50 µm (F) Percentage of total DAPI cells labeled with KI67 in control and mutant mouse cortices. All data points represent mean, n = 30 brain sections for each genotype, from three independent experiments. Student's t-test, ***p<0.001. (G) Confocal images of coronal section from E10.5-E14.5 control and mutant mouse cortices stained with apoptosis marker activated caspase-3 (CC3; red) and counterstained with DAPI (blue). Scale bar, 20 um for all groups. (H) Percentage of DAPI cells that are also caspase3+ from E10.5-E14.5. All data points represent mean ± SEM, n = 35–46 brain sections for each genotype, from three independent experiments. Two-way ANOVA with Sidak's multiple comparison test; within each age group, +/+ was compared to ΔN/ΔN; *p<0.05, **p<0.01, ns, not significant. Detailed tabulation of means, SEMs, sample sizes, and exact p-values can be found in *Figure 3—source data 1*.

The online version of this article includes the following source data and figure supplement(s) for figure 3:

**Source data 1.** Mean, SEM, sample size (n), and exact p-values for *Figure 3* quantifications.
**Figure supplement 1.** Intact organization of apical complex proteins and skewed cleavage plane orientation in *Ocln*-mutant mouse.
**Figure supplement 1—source data 1.** Mean, SEM, sample size (n), and exact p-values for *Figure 3—figure supplement 1* quantifications.

---

Prolonged mitosis has been shown to impair self-renewal of RGCs, leading to increases in intermediate progenitor cells (IPCs), neurons, and/or apoptotic events (*Pilaz et al., 2016*). Both E10.5 and E12.5 cortices, but not E14.5 cortices, revealed a higher percentage of activated (cleaved) caspase 3 (CC3)-positive apoptotic cells in mutant embryos compared to controls (*Figure 3G,H*). Thus, delayed mitosis leads to more apoptotic events that are concurrent with time points (E10.5 and E12.5) in which mOCLN-FL should be expressed but is lost in *Ocln*-mutants. Similar to OCLN knockdown in MDCK cells that impaired mitotic cleavage angle of those dividing kidney epithelial cells (*Odenwald et al., 2017*), we also observed skewed cleavage plane orientation (*Pilaz et al., 2016*; *Jayaraman et al., 2016*) in mutant RGCs compared to control (*Figure 3—figure supplement 1B,C*). Taken together, these results suggest that loss of mOCLN-FL expression impacts mitotic behavior, lengthening M-phase, and increasing apoptotic events prior to E14.5.

### *Ocln*-mutant mice display precocious neuronal differentiation, progenitor pool depletion and thinning cortex

Based on our observation of reduced progenitor proliferative capacity in E12.5 *Ocln*-mutant cortices (*Figure 3E,F*) and prolonged M-phase, we hypothesized that *Ocln* mutation would also increase the TUJ1+ neuronal cell population. At E12.5, when the rate of neurogenesis normally rises (*Haubensak et al., 2004*; *Huttner and Kosodo, 2005*; *Mione et al., 1997*), we observed a further increase in TUJ1+ neurons and a reduction of PAX6+ RGCs in *Ocln*$^{ΔN/ΔN}$ compared to wild-type mice (*Figure 4A,B*). Furthermore, the percentage of cycling intermediate progenitor cells (IPCs) dual-labeled with TBR2 and KI67 antibodies is greater in mutant cortices, suggesting both the rates of symmetric neurogenic divisions and indirect neurogenic (IPC-generating) RGC divisions are increased at the expense of the RGC progenitor pool (*Figure 4C,D*). Dual-staining of TBR2 and proliferation marker KI67 ensured that cycling IPCs were selected and not TBR2+ post-mitotic neurons. Later in corticogenesis, although the TUJ1+ cell layer was thicker in mutant compared to control cortices at E12.5, by E16.5 *Ocln* mutant cortices displayed thinner TUJ1+ cortical layers, indicating that precocious differentiation of neural progenitors at E12.5 leads to thinning of mutant cortex at later ages (*Figure 4E,F*). Finally, we analyzed postnatal day 1 (P1) cortices for layer markers and found a significant reduction upon immunolabeling of both deep-layer CTIP2+ and superficial-layer CUX1+ cells in *Ocln*$^{ΔN/ΔN}$ cortex (*Figure 4G,H*). Together, these data indicate that loss of mOCLN-FL in early corticogenesis leads to precocious neuronal differentiation, reducing cell output, contributing to cortex thinning by E16.5.

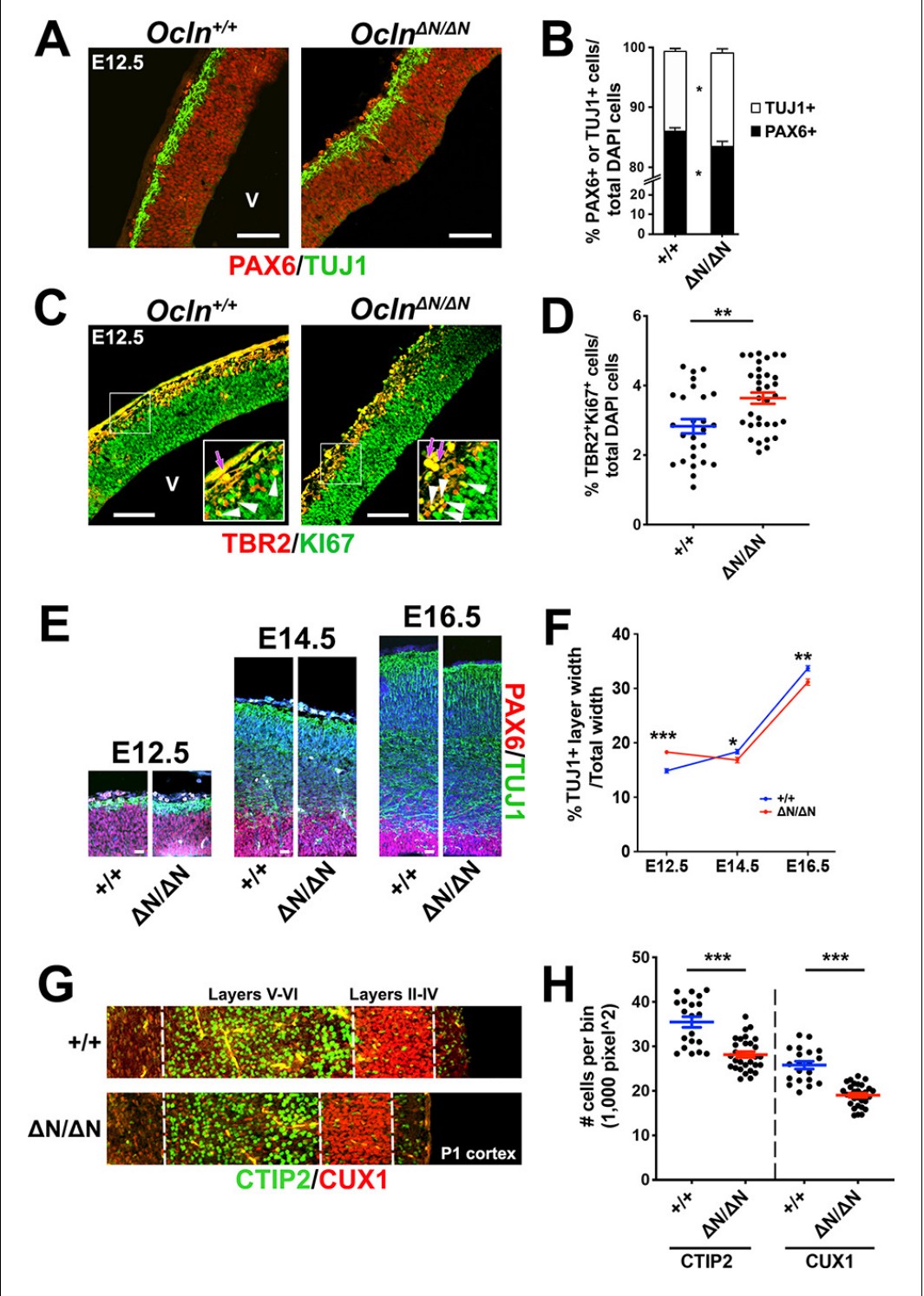

**Figure 4.** Progenitor depletion, precocious neuronal differentiation, and microcephaly in *Ocln*[ΔN/ΔN] cortex. (**A**) Confocal images of coronal sections from E12.5 control and mutant mouse cortices stained with RGC marker PAX6 (red) and neuronal marker TUJ1 (green). V, ventricle. Scale bar, 50 um. (**B**) Quantification of TUJ1+ or PAX6+ cell percentages in control and mutant mice. All data points represent mean ± SEM, n = 26 brain sections from three independent experiments for each genotype. Student's t-test, *p<0.05. (**C**) Confocal images of coronal section from E12.5 control and mutant mouse cortices stained with intermediate progenitor marker TBR2 (red) and proliferation marker KI67 (green). Inset enlargements show regions outlined by white box. Dual-stained TBR2+/Ki67+ yellow IPCs (white arrowhead) were quantified while auto-fluorescent red blood cells (purple arrows) were

*Figure 4 continued on next page*

*Figure 4 continued*

disregarded in quantification. V, ventricle. Scale bar, 50 μm. (**D**) Quantification of dual-stained TBR2+ and KI67+ cell percentages in control and mutant mice. All data points represent mean ± SEM, n = 26 sections for $Ocln^{+/+}$, n = 33 sections for $Ocln^{\Delta N/\Delta N}$, from three independent experiments for each genotype. Student's t-test, \*\*p<0.01. (**E**) Confocal microscope images of E12.5-E16.5 wild-type and mutant mouse cortices stained with PAX6 (red) and TUJ1 (green) and counterstained with DAPI (blue). Scale bar, 20 um (**F**) Time-course quantification of TUJ1+ layer width of E12.5 to E16.5 mouse cortices. All data points represent mean ± SEM, n = 34 brain sections for E12.5 $Ocln^{+/+}$, n = 48 for E12.5 $Ocln^{\Delta N/\Delta N}$, n = 41 for E14.5 $Ocln^{+/+}$, n = 51 for E14.5 $Ocln^{\Delta N/\Delta N}$, n = 28 for E16.5 $Ocln^{+/+}$, and n = 30 for E16.5 $Ocln^{\Delta N/\Delta N}$, from three independent experiments for each age and each genotype. Two-way ANOVA with Sidak's multiple comparison test was performed comparing control and mutant in each age group, \*p<0.05; \*\*p<0.01; \*\*\*p<0.001. (**G**) Confocal images of P1 wild type and mutant mice cortices stained for CTIP2 and CUX1 to label deep layers (**V and VI**) and upper layers (**II-IV**), respectively. (**H**) Quantification of Ctip2+ or Cux1 + cell numbers per bin. All data points represent mean ± SEM, n = 20 for $Ocln^{+/+}$ and n = 30 for $Ocln^{\Delta N/\Delta N}$ from three independent experiments for each genotype. Student's t-test, \*\*\*p<0.001. Detailed tabulation of all means, SEMs, sample sizes, and exact p-values can be found in *Figure 4—source data 1*.

The online version of this article includes the following source data for figure 4:

**Source data 1.** Mean, SEM, sample size (n), and exact p-values for *Figure 4* quantifications.

## Generation of OCLN-hESC lines modeling $Ocln^{\Delta N/\Delta N}$ mice and patient truncations

Mouse models of microcephaly have been crucial in elucidating potential mechanisms of disease progression. However, differences in their progenitor characteristics, including the abundance of outer radial glial cells (oRG) in humans limits the ability of mouse models to recapitulate all aspects of human brain malformations (*Lizarraga et al., 2010*; *Barrera et al., 2010*; *Pulvers et al., 2010*; *Gruber et al., 2011*). Human embryonic stem cells (hESCs) offer advantages for in vitro 3D culture models of cytoarchitectural events in human cortical development and can facilitate the exploration of disease pathogenesis [reviewed in 41]. CRISPR/Cas9 genome editing was employed to interrupt *OCLN* in hESCs at exon 3, the same exon targeted in the $Ocln^{\Delta N/\Delta N}$ mutant mouse (*Saitou et al., 2000*) and the location of half of the reported *OCLN* truncating mutations in patients (*Abdel-Hamid et al., 2017*; *O'Driscoll et al., 2010*; *Jenkinson et al., 2018*; *Aggarwal et al., 2016*; *Elsaid et al., 2014*). The human *OCLN* gene was targeted at two separate loci within exon 3 using distinct sgRNAs to create two mutant lines, termed sg5 and sg11 (*Figure 5A*, *Figure 5—figure supplement 1*). Six human OCLN (hOCLN) isoforms have been discovered to date (*Kohaar et al., 2010*), compared to two mOCLN isoforms, although only four isoforms have confirmed expression in the human brain. Our study focused on three of these isoforms that displayed the smallest inter-sample variation in expression levels among all other isoforms. CRISPR/Cas9 targeting of exon 3 resulted in lost expression of all explored hOCLN isoforms except for hOCLN-ex3del, as this isoform skips exon 3 (*Figure 5B*). A list of potential off-target sites was obtained through http://chopchop.cbu.uib.no/ and each locus was sequenced in both wild-type and mutant cell lines. No off-target mutations were observed in either sg5 or sg11 mutant hESC lines (*Figure 5—figure supplement 2*). Thus, our hESC sg5 and sg11 mutant lines closely resemble the $Ocln^{\Delta N/\Delta N}$ mouse mutant and can serve as useful tools in elucidating human phenotypes of OCLN deficiency.

## OCLN colocalizes with centrosomes in vitro in hESCs

As in the mouse, full-length hOCLN (hOCLN-wt) can be found in both membrane and soluble nuclear fractions (*Figure 5C*). Endogenous hOCLN-ex4del, which is missing exon four and consequently lacks the fourth of its transmembrane domains, also localizes to both membrane and soluble fractions, while the hOCLN-ex3del splice form, missing three of its four transmembrane domains, was found only in the soluble nuclear fraction. Immunocytochemical analysis using an N-terminus-specific OCLN antibody confirmed that both sg5 and sg11 undifferentiated mutant lines have lost expression of hOCLN-wt and hOCLN-ex4del, both normally expressed at the plasma membrane and at the centrosome (*Figure 5D*). However, hOCLN-ex3del localized solely at the centrosome and remained expressed in sg5 and sg11 mutant lines as shown by immunoreactivity with C-terminus OCLN antibody (*Figure 5E*). These results are consistent with OCLN staining observed in the mouse cortex (*Figure 2*).

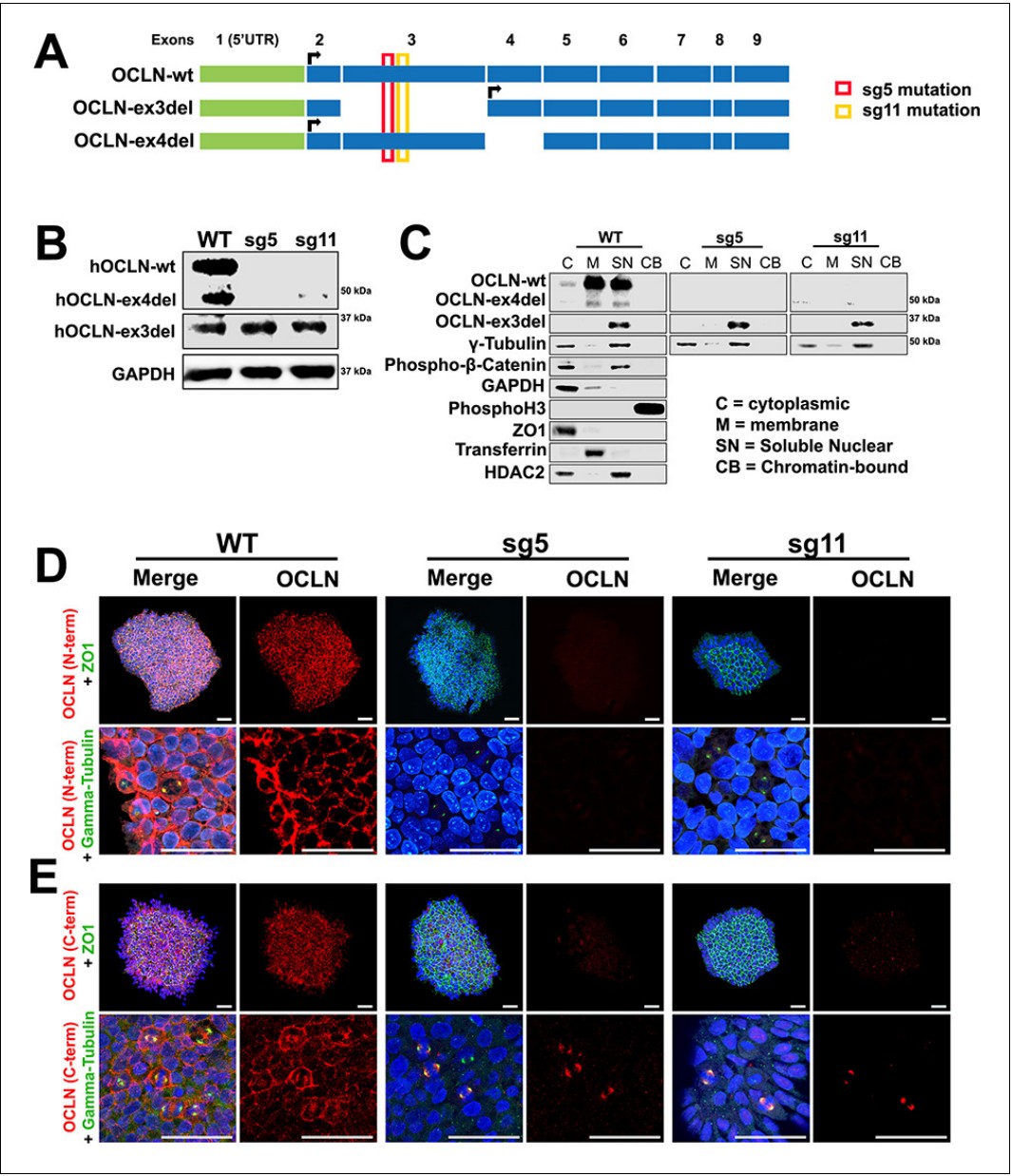

**Figure 5.** Human full-length and truncated OCLN isoform expression in WT and genome-edited embryonic stem cells. (**A**) Human *OCLN* transcripts explored in this study. Shown are three OCLN isoforms expressed the human brain and that exhibited the lowest intersample variation amongst human liver samples (***Kohaar et al., 2010***). The location of Cas9/CRISPR targeting is denoted in red and yellow for the sg5 and sg11 mutants, respectively. (**B**) Western blot analysis of hESC WT, sg5, and sg11 lysates. Representative image is from four independent replicates. GAPDH is used as loading control. (**C**) Subcellular localization of OCLN by immunoblot analysis of subcellular fractions (cytoplasmic, membrane, soluble nuclear, and chromatin-bound) in WT, sg5, and sg11 hESC lines. Centrosomal markers γ-tubulin and phospho-β-catenin colocalize with OCLN in the soluble nuclear fraction. Fraction-specific controls used are GAPDH and ZO1 (cytoplasmic), HDAC2 (soluble nuclear), Transferrin (membrane), Phospho-histone3 (chromatin-bound). (**D and E**) Confocal microscope images of hESC colonies from WT, sg5, and sg11 cultures stained with N-terminus-specific (**D**) and C-terminus-specific (**E**) OCLN antibodies and either tight junction marker ZO1 or centrosomal marker γ-tubulin. Nuclei are stained with DAPI. Scale bar, 50 um. The online version of this article includes the following figure supplement(s) for figure 5:

**Figure supplement 1.** CRISPR mutagenesis targets OCLN at two separate loci on exon 3.
**Figure supplement 2.** Off-target analysis of sg5 and sg11 *OCLN* mutant lines.

## OCLN-deficient hESC-derived cortical organoids exhibit reduced size, increased mitotic index, apoptosis, and premature neuronal differentiation

To assess the pathogenesis of human cortical phenotypes associated with loss of full-length *OCLN*, we generated 3D cortex-like organoids (aka, cortical spheroids) from our control and gene-edited hESC lines using a modification of an established protocol (*Paşca et al., 2015*; *Lee et al., 2017*). Organoids were examined for the presence of forebrain-specific markers PAX6, FOXG1, and OTX1 and lack of midbrain and hindbrain markers DLX1 or NKX2.1 (*Figure 6—figure supplement 1A*). Furthermore, we probed organoids for OCLN expression after 20 days of differentiation (d20) and observed membrane and centrosome localization, similar to OCLN localization in the mouse cortex (*Figure 6—figure supplement 1B*). We also saw ~40% and~60% reductions in the circumferences of mutant organoids at d20 and d40, respectively (*Figure 6A,B*). To explore the mechanisms underlying the size reduction, we assessed the proliferative capacity of d10 organoids and found reduced proliferation and an increased mitotic index in sg5 and sg11 mutant organoids (*Figure 6C–E*). By d20, mutant lines exhibited increased apoptosis and increased TUJ1+ neuronal cell populations compared to controls (*Figure 6F–I*). These results suggest that hOCLN-wt deficiency in this 3D human in vitro model mimics the mouse model with respect to prolonged M-phase, premature neuronal differentiation, and increased apoptosis.

A type of basal neural progenitor, the oRG, is abundant in humans and non-human primates but scarce in rodents (*Betizeau et al., 2013*; *Wang et al., 2011*; *Hansen et al., 2010*; *Fietz et al., 2010*). The oRG population is largely responsible for the expansion of human cerebral cortex [reviewed in 48]. We therefore examined organoids by immunostaining for oRG marker, HOPX, compared to early neuronal marker NeuroD1. The ratio of HOPX+/NeuroD1+ cell numbers was greatly reduced in the mutant organoids (*Figure 6J,K*). This suggests that human OCLN is important for the balance between oRGs and neurons in the human cortex and may explain the more pronounced deficit in human organoid size compared to the mild reduction in cortical size of the mouse.

## OCLN interacts with mitotic spindle proteins RAN, NuMA and mediates spindle integrity

Based on published proteomic data in MDCK cells (*Fredriksson et al., 2015*), we prioritized potential OCLN interacting partners for co-immunoprecipitation-Western blot analysis. Among these candidates were small GTPase nuclear protein RAN, known for its role in nucleo-cytoplasmic transport and spindle assembly (*Cavazza and Vernos, 2015*; *Carazo-Salas et al., 2001*; *Caudron et al., 2005*; *Li et al., 2007*) and NuMA, a key player in spindle assembly and maintenance (*Chu et al., 2016*; *Silk et al., 2009*). We observed a robust pulldown of both hOCLN-wt and hOCLN-ex3del isoforms by anti-NuMA antibody and reciprocal pulldown of NuMA by C-terminus OCLN antibody (thus recognizing both hOCLN-wt and hOCLN-ex3del isoforms) from hESC lysates (*Figure 7A*). Similar pulldown results were obtained with anti-RAN antibody and reciprocal pulldown of RAN with anti-OCLN antibody (*Figure 7—figure supplement 1*). NuMA-OCLN interaction was further supported by immunostaining in which co-localization of both proteins was observed at the mitotic spindle poles during M-phase (*Figure 7B*). NuMA-labeled spindle pole morphology appeared rounded more frequently in sg5 and sg11 mutants (*Figure 7C–F*), specifically exhibiting a circular shape rather than the expected elongated or half-moon shape of normal NuMA immunostaining (*Nguyen and Munger, 2009*). Since NuMA is known to be important to mitotic spindle stability and chromosome alignment (*Chu et al., 2016*; *Silk et al., 2009*) and has also been shown to regulate astral spindle orientation and tethering to the cell cortex (*Seldin et al., 2016*), we explored astral spindle morphology and showed that both sg5 and sg11 mutant lines exhibited significantly weaker astral spindle labeling of alpha-tubulin, with shorter and fewer spindles, compared to control (*Figure 7G,H*). Finally, we observed a higher frequency of altered mitotic spindle features such as abnormal spindle tilts and asymmetric spindle densities, higher frequency of chromosome misalignment, and reduced intensity of acetylated alpha-tubulin staining, in sg5 and sg11 mutants compared to control (*Figure 7I–M*). Chromosome misalignments as a result of impaired spindle integrity will likely result in aneuploidy in mutant cells. The karyotype of wild-type undifferentiated hESCs was indistinguishable from that of undifferentiated hESC-sg5 or hESC-sg11 cells. However, the regulation of spindle pole and cleavage plane orientation is especially critical for cortical radial glia making symmetric

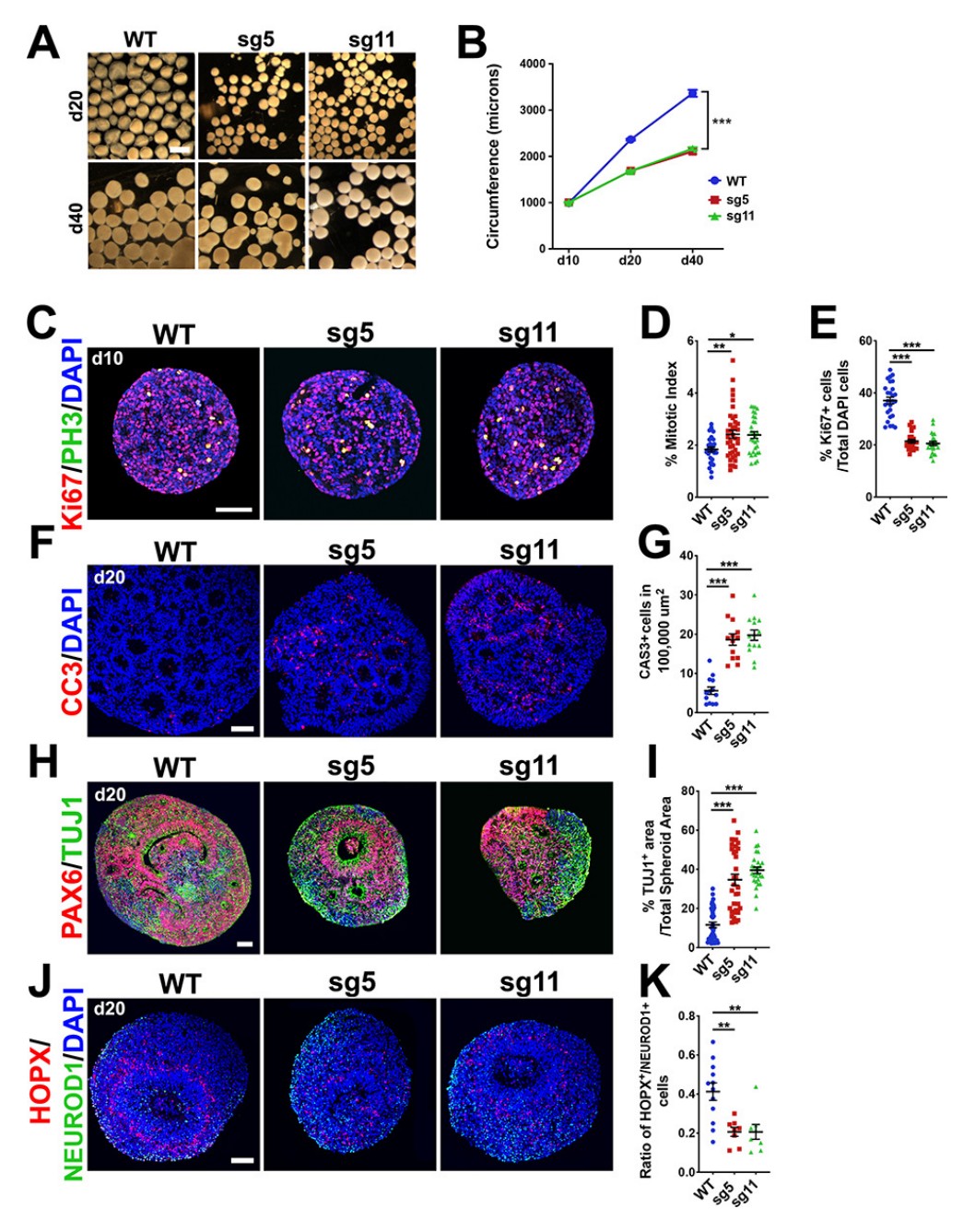

**Figure 6.** Progenitor proliferation defects, precocious neuronal differentiation and apoptosis in 3-D cortical spheroids from *OCLN* sg5 and sg11 hESC mutants. (**A**) Brightfield images of d20 and d40 organoids derived from WT, sg5, and sg11 hESC cultures. Scale bar, 1 mm. (**B**) Time-course of d10, d20, and d40 spheroid circumference (in microns). All data points represent mean ± SEM, n = 61 d10 WT organoids, n = 67 d10 sg5 organoids, n = 40 d10 sg11 organoids, n = 246 d20 WT, n = 170 d20 sg5 organoids, n = 208 d20 sg11 organoids, n = 91 d40 WT organoids, n = 98 d40 sg5 organoids, and n = 50 d40 sg11 organoids, collected from five independent experiments for each genotype and each age.. Two-way ANOVA was performed with Tukey's multiple comparison test, comparing all three lines to each other within each age group, ***p<0.001. (**C**) Confocal microscope images of 10-micron sections from d10 WT, sg5, and sg11 spheroids stained with proliferation marker KI67 (red), mitotic marker PH3 (green) and counterstained with DAPI (blue). Scale bar, 50 um. (**D**) Mitotic index, defined as the percentage of proliferating, KI67+ cells co-labeled with PH3 in WT, sg5, and sg11 individual organoids. All data points represent mean ± SEM, n = 32 WT, 39 sg5, and 30 sg11 full spheroids, from three independent experiments. One-way ANOVA with Tukey's multiple comparison test, **p<0.01, *p<0.05. (**E**) Percentage of DAPI

*Figure 6 continued on next page*

*Figure 6 continued*

labeled cells that are KI67+ in WT, sg5, and sg11 individual organoids. All data points represent mean ± SEM, n = 26 WT, n = 23 sg5, and n = 23 sg11 organoids, from three independent experiments. One-way ANOVA with Tukey's multiple comparison test, ***p<0.001. (F) Confocal images of 10-micron sections from d20 WT, sg5, and sg11 organoids stained with apoptosis marker activated caspase-3 (red) and counterstained with DAPI (blue). Scale bar, 50 µm. (G) Percentage of DAPI labeled cells that are caspase3+. All data points represent mean ± SEM, n = 13–14 organoids for each genotype, from two independent experiments. One-way ANOVA with Tukey's multiple comparison test, ***p<0.001. (H) Confocal images of 10-micron sections from d20 WT, sg5, and sg11 organoids stained with RGC marker PAX6 (red), neuronal marker TUJ1 (green) and counterstained with DAPI (blue). Scale bar, 50 µm. (I) Quantification of TUJ1+ area per total organoid area or PAX6+ cell percentages in WT, sg5, and sg11 organoids. All data points represent mean ± SEM, n = 38 organoids for WT, n = 34 for sg5, and n = 27 for sg11, from four independent experiments for each group. One-way ANOVA with Tukey's multiple comparison test, ***p<0.001. (J) Confocal images of 10-micron sections from d20 WT, sg5, and sg11 organoids stained with oRG marker HOPX (red), neuronal marker NeuroD1 (green) and counterstained with DAPI (blue). Scale bar, 50 µm. (K) Ratio of HOPX+ cells to NeuroD1+ cells in WT, sg5, and sg11 individual organoids. All data points represent mean ± SEM, n = 12 WT organoids, n = 8 sg5 organoids, and n = 8 sg11 organoids, from two independent experiments for each group. One-way ANOVA with Tukey's multiple comparison test, **p<0.01. Detailed tabulation of all means, SEMs, sample sizes, and exact p-values can be found in *Figure 6—source data 1*.

The online version of this article includes the following source data and figure supplement(s) for figure 6:

**Source data 1.** Mean, SEM, sample size (n), and exact p-values for *Figure 6* quantifications.
**Figure supplement 1.** Human Cortical Spheroid (hCS) characterization and OCLN localization.

---

stem divisions (*Huttner and Kosodo, 2005*). We therefore compared the karyotypes of metaphase cells from dissociated d20 WT, sg5, and sg11 3-D organoids (*Figure 7—figure supplement 2*). The frequency of one or more abnormal chromosomes per cell was significantly increased in sg5 and sg11 organoids compared to control organoids. This would be expected to contribute to the observed increase in apoptosis and is consistent with the prolonged M-phase and cell death seen at E10.5 and E12.5 in the $Ocln^{\Delta N/\Delta N}$ mouse. Together, these results provide compelling evidence for OCLN mediation of spindle integrity and stability.

## Discussion

Here, we report a previously unappreciated role of tight junction protein OCLN in neurogenesis, in both mouse in vivo and human in vitro models. We show that OCLN is located in the embryonic cortical epithelium at tight junctions in the plasma membrane and at the centrosome through the NE-to-RGC transition and only a truncated OCLN splice form is located at the centrosome thereafter. Furthermore, loss of full-length OCLN impairs cortical neurogenesis in both hESC-derived cortical organoids and mutant mouse cortex, recapitulating *OCLN* mutations in humans. Specifically, we show that loss of full-length OCLN in the mouse embryonic cortex leads to microcephaly due to prolonged M-phase, a transient burst of apoptosis, and precocious neuronal differentiation at the expense of the progenitor pool. A more pronounced size deficit was observed in *OCLN* mutant hESC-derived cortical organoids, associated with reduced proliferation, premature differentiation, apoptosis, and increased aneuploidy. Finally, OCLN interacts with mitotic spindle protein NuMA, known to maintain mitotic spindle stability and integrity. Loss of several OCLN isoforms in hESCs produced altered spindle pole morphology, fewer and shorter astral microtubules, and impaired mitotic spindle integrity, indicating apparatus malfunction, as summarized in *Figure 8*. We previously showed cell cycle perturbations that disrupt the balance between RGC and intermediate progenitor cell (IPC) neural populations are associated with PMG (*Mirzaa et al., 2014*) and an analogous imbalance is proposed to underlie PMG in this *OCLN* phenotype.

Half of reported human *OCLN* mutations are located on exon 3 and are protein-truncating (*Abdel-Hamid et al., 2017*; *O'Driscoll et al., 2010*; *Jenkinson et al., 2018*; *Aggarwal et al., 2016*; *Elsaid et al., 2014*), likely resulting in loss-of-function of full-length OCLN. One might consider that the observed effects on mitotic spindles and impaired cortical neurogenesis are caused instead by a dominant negative effect triggered by the presence of truncated OCLN isoforms. However, a number of features argue against this possibility. First, this human disorder is caused by autosomal

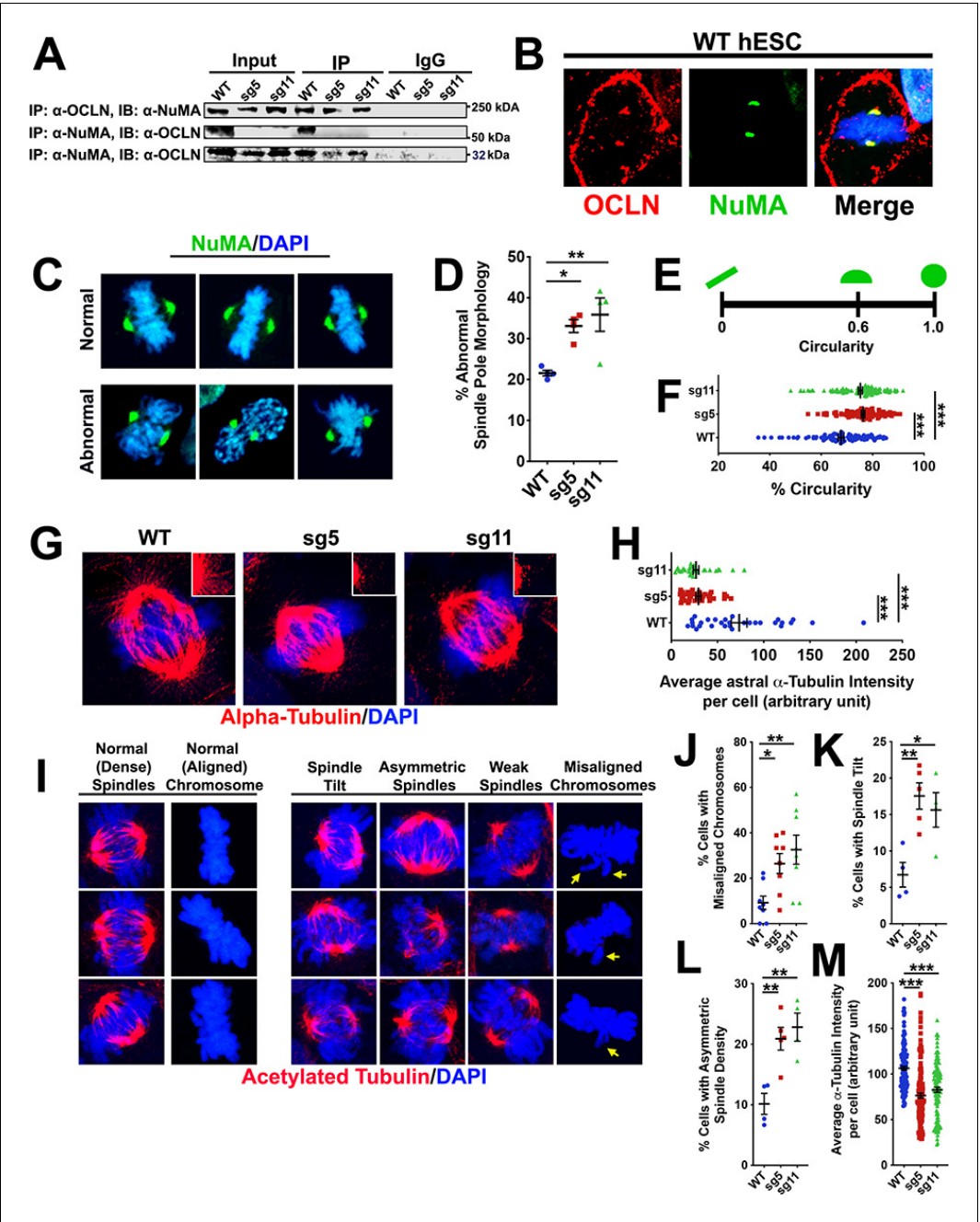

**Figure 7.** Full-length OCLN deficit impairs hESC mitotic spindle and astral microtubule organization and integrity. (**A**) Immunoprecipitation (IP) of endogenous OCLN protein with anti-OCLN antibody and of endogenous NuMA protein with anti-NuMA from hESCs lysates. IP = antibody used to pulldown protein of interest and its adjacent interaction partners; immunoblot (IB) = antibody used to probe for presence of interaction partners. The input represents about 3% of total lysate. Non-specific IgG used as negative control. (**B**) Confocal images of WT hESCs stained with anti-NuMA (green) and anti-OCLN (red) antibodies. DAPI (blue) stains nuclei. (**C**) Representative confocal images of normal and abnormal NuMA-labeled spindle pole morphology in green and counterstained with DAPI (blue). (**D**) Percentage of cells with abnormal spindle pole morphology in WT, sg5, and sg11 culture. All data points represent mean ± SEM, n = 4 independent experiments for each group, with each experiment analyzing 15 cells from that group. One-way ANOVA with Tukey's multiple comparison test, *p<0.05, **p<0.01. (**E, F**) Schematic of circularity metric in **E**) used to analyze the percent circularity of each spindle pole in WT, sg5, sg11. Quantification of % circularity in **F**) of WT, sg5, sg11 culture. All data points ± SEM, n = 92–115 for all groups, from four independent experiments. One-way ANOVA with Tukey's multiple comparison test, ***p<0.001. (**G**) Confocal images of astral spindle as stained with alpha-tubulin (red) and counterstained with DAPI (blue). (**H**) Quantification

*Figure 7 continued on next page*

*Figure 7 continued*

of average alpha-tubulin fluorescent intensity specifically of astral spindles in WT, sg5, and sg11 culture. All data points represent mean ± SEM, n = 30 cells for all groups, from three independent experiments for each group. One-way ANOVA with Tukey's multiple comparison test, ***p<0.001. (I) Representative confocal images of hESCs stained with acetylated alpha-tubulin (red) and nuclear stain DAPI (blue), grouped by their mitotic spindle structure, density, and orientation. (J–L) Percentage of cells exhibiting misaligned chromosomes (J), spindle tilts (K), or asymmetric spindle density (L) in WT, sg5, and sg11 culture. All data points represent mean ± SEM. For spindle tilt and asymmetric spindle densities, n = 4–5 independent experiments, with each experiment analyzing 15 cells each, for all groups. For chromosome misalignment, n = 8 independent experiments for all groups, with each experiment analyzing 15 cells. One-way ANOVA with Tukey's multiple comparison test, *p<0.05, **p<0.01. (M) Quantification of relative alpha-tubulin fluorescent intensity in WT, sg5, and sg11 culture. All data points represent mean ± SEM, n = 128 cells for WT, n = 137 cells for sg5, and n = 116 cells for sg11, from four independent experiments for each group. One-way ANOVA with Tukey's multiple comparison test, ***p<0.001. Detailed tabulation of all means, SEMs, sample sizes, and exact p-values can be found in *Figure 7—source data 1*.

The online version of this article includes the following source data and figure supplement(s) for figure 7:

**Source data 1.** Mean, SEM, sample size (n), and exact p-values for *Figure 7* quantifications.
**Figure supplement 1.** OCLN interacts with mitotic spindle protein RAN.
**Figure supplement 2.** Mutant hESC-derived cortical organoid neural progenitors exhibit increased aneuploidy.

recessive mutation, which speaks against a dominant negative effect since heterozygous individuals are asymptomatic. Second, a number of reported insertion/deletion *OCLN* mutations causing BLC-PMG were found on exons 5, 6, or 7, which would result in loss-of-function of all OCLN isoforms (*Abdel-Hamid et al., 2017*; *O'Driscoll et al., 2010*; *Jenkinson et al., 2018*; *Aggarwal et al., 2016*; *Elsaid et al., 2014*). Third, there is no indication yet to suggest a genotype-phenotype correlation with clinical severity between the more proximal vs. distal *OCLN* mutations. Together these features strongly indicate that loss-of-function of full-length OCLN is central to the phenotypes observed in this study. What remains to be understood is why expression of the truncated isoform at the centrosome continues to be required after the developmental down regulation of the full-length protein under physiological conditions.

We utilized a previously characterized *Ocln* mutant mouse model presumed to be an *Ocln*-null (*Saitou et al., 2000*). While numerous in vitro studies suggest an important role of occludin in formation or stabilization of the TJ paracellular barrier (*Furuse et al., 1996*; *Balda, 1996*; *Van Itallie and Anderson, 1997*; *McCarthy et al., 1996*), the *Ocln* mutant mice did not exhibit gross phenotypes that might be expected of impaired TJs (*Saitou et al., 2000*; *Schulzke et al., 2005*). Thus, the exact contributions of occludin to TJ formation and maintenance have remained controversial. However, our observation of continued mOCLN-ΔN expression suggests the *Ocln* mutant mouse is actually a hypomorph. Previous studies have shown the OCLN C-terminus to be important for ZO-1 interaction, proper localization to the TJ, and other signaling functions (*Furuse, 1994*; *Tash et al., 2012*). Therefore, the mOCLN-ΔN isoform which still contains the long C-terminal tail and one of four transmembrane domains (*Figures 1* and *2*) may be poised to contribute to TJ formation or stability and warrants further study.

Our finding OCLN at the RGC centrosome is consistent with recent studies in which endogenous OCLN localized to the centrosome in endothelial cell culture, in mouse retinal vasculature, and in surgical samples of retinal neovascular endothelium (*Liu et al., 2016*). Exogenously expressed full-length occludin also localized to centrosomes in MDCK cells in vitro and was further shown to regulate mitotic entry and centrosome separation in a phospho-dependent manner (*Runkle et al., 2011*). Here, both full-length and truncated isoforms localized to the cortical RGC centrosome (*Figures 1*, *2* and *5*), suggesting the C-terminal domain of OCLN is important for this localization. This hypothesis is consistent with previous demonstration that OCLN phosphosite S490, located at the distal end of the C-terminus, is important for centrosome localization (*Runkle et al., 2011*). In contrast, neither mouse mOCLN-ΔN nor human hOCLN-ex3del localized to cortical progenitor plasma membranes, likely due to the loss of three of four transmembrane domains in the truncated isoform that lacks a signal peptide for membrane insertion. Our data strongly supports the notion that OCLN has two distinct and role-dependent subcellular localizations. This study also expands the functional

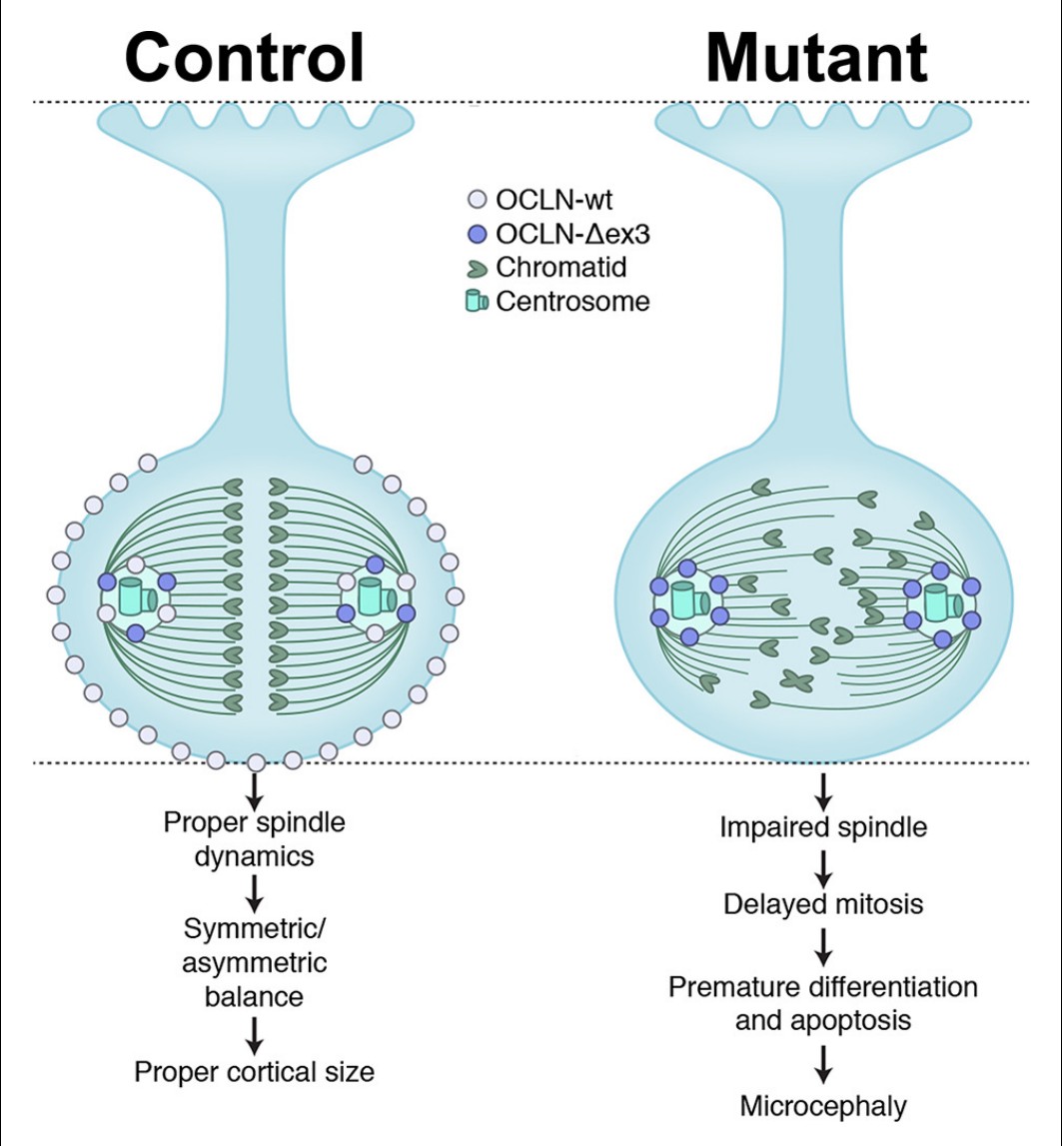

**Figure 8.** Summary model of OCLN isoform functions in radial glial cells (RGCs) explored in this study.

repertoire of OCLN to regulation of neural proliferation and brain size. Although OCLN has been previously linked to epithelial and endothelial cell proliferation in vitro (*Bolinger et al., 2016*; *Runkle et al., 2011*) and mouse retinal neovascularization and angiogenesis in vivo (*Liu et al., 2016*), ours is the first recognition of its role in cortical progenitor proliferation.

While the presence of mOCLN-ΔN/hOCLN-ex3del isoforms exclusively at the centrosome imply an interesting role in proliferation, the present data particularly emphasize the role of full- length OCLN in mitotic regulation. First, loss of mOCLN-FL during E10.5-E12.5 coincided with aberrant prolonged M-phase and a transient burst of apoptosis, neither of which continued in control or mutant mice beyond the age when OCLN-FL is normally absent. Second, selective loss of mOCLN-FL produced microcephaly in postnatal mice, even with the continued expression of mOCLN-ΔN. Third, both full-length and truncated isoforms bind to NuMA and RAN, proteins important for spindle pole morphology and spindle integrity, structures that are altered by OCLN deficit. Together, these data provide compelling evidence for a role of full-length OCLN in mitotic spindle regulation and mitotic progression at a crucial stage in neural progenitor proliferation. Our data suggest that there is a 'critical period' of mOCLN-FL function in the developing cortical neuroepithelium and lasts through E12.5, spanning the period of the NE-to-RGC transition. The accrued reduction of the

progenitor pool, by precocious differentiation and increased apoptosis, severely impacts subsequent neuronal differentiation, leading to cortex thinning by E16.5 and culminating in microcephaly by P7 and P20.

Faint, diffuse nuclear staining of OCLN was observed in some, but not all, of $Ocln^{\Delta N/\Delta N}$ neural progenitors (*Figure 2C*). While this may represent non-specific antibody labeling, a role for one or more OCLN isoforms in the nucleus cannot be ruled out at this time. Indeed, TJ proteins ZO-1 and ZO-2 were found to localize to the nucleus in vitro in a cell-confluency-dependent manner (*Gottardi et al., 1996*; *Jaramillo et al., 2004*; *Traweger et al., 2003*). Furthermore, it is interesting that previous proteomic studies probing for OCLN-associated proteins found that nuclear envelope protein Emerin pulled-down with OCLN in vitro (*Fredriksson et al., 2015*). Clarification of a possible nuclear function for OCLN awaits further study.

One prominent mechanistic model of microcephaly proposes that prolonged mitosis of ventricular neural progenitors will alter division outcomes of RGCs, promoting neuronal differentiation and apoptosis at the expense of the progenitor pool and so lead to microcephaly (*Pilaz et al., 2016*; *Lizarraga et al., 2010*; *Chen et al., 2014*). Consistent with this hypothesis, our results indicate that mitosis is prolonged in both mouse $Ocln^{\Delta N/\Delta N}$ cortex and human 3-D organoids from two independent (sg5 and sg11) genome-edited hESC lines (*Figures 4* and *6*) and parallels the human microcephaly phenotype. Similar to the mouse data, loss of the hOCLN-wt, hOCLN-ex4del, and hOCLN-ex7ext isoforms in 3-D cultures resulted in premature neuronal differentiation at the expense of the progenitor pool. Unlike the mouse, HOPX$^+$ oRGs constitute the majority of the human neural progenitor population (*LaMonica et al., 2012*) and these were severely affected by OCLN loss in 3-D culture. The paucity of oRG cells in rodents likely contributes to the significant but milder degree of microcephaly in the mouse compared to human patients, who typically display occipital-frontal head circumferences at ~4 standard deviations below the mean or smaller. Importantly, since organoids lack vasculature and inflammatory cells, the significant impact of these OCLN mutations on mitotic spindle integrity, proliferation and survival are shown to be independent of potential deleterious effects from impaired vascular neogenesis, perivascular extravasation and calcification. Finally, our results link OCLN function with mitotic and astral spindle integrity, likely in part through its interactions with proteins like NuMA and RAN. In the present study, astral spindle abnormalities associated with loss of full-length OCLN are consistent with an earlier observation that perturbations in astral spindles by disruption of LGN limited their reach to the apical and basal regions of the cell's cortical actin meshwork, skewed the cleavage plane and increased the rate of asymmetric, neurogenic divisions of cortical progenitors (*Mora-Bermúdez et al., 2014*). Spindle protein LGN is a well-known interaction partner of NuMA (*Du et al., 2001*; *Morin et al., 2007*), which we show interacts with OCLN in neural progenitors. Together, these data illuminate a new role for OCLN in neurogenesis and survival with a specific requirement for full-length OCLN in the promotion of early progenitor self-renewal through proper neural stem cell mitotic spindle function.

## Materials and methods

### Key resources table

| Reagent type (species) or resource | Designation | Source or reference | Identifiers | Additional information | RRID |
|---|---|---|---|---|---|
| Cell line (*Homo-sapiens*) | Human embryonic stem cell | WiCell | WAe009-A | NIH registry #0062 | RRID:CVCL_9773 |
| Genetic reagent (mouse) | *Ocln$^{\Delta N/+}$* (*previously Ocln$^{+/-}$*) | *Saitou et al., 2000* DOI: 10.1091/ mbc.11.12.4131 | C57BL/6 background | Gift of Dr. Margaret Neville (University of Colorado, Denver) | RRID:MGI:3716350 |

*Continued on next page*

Continued

| Reagent type (species) or resource | Designation | Source or reference | Identifiers | Additional information | RRID |
|---|---|---|---|---|---|
| Antibody | Anti-Occludin (Rb polyclonal) | Abcam | Cat# ab31721 | IHC(1:4000)* IF(1:4000)* WB(1:2000) *Tyramide signal amplification used | RRID:AB_881773 |
| Antibody | Anti-Occludin (Ms monoclonal) | BD Biosciences | Cat# 611091 | IHC(1:4000)* *Tyramide signal amplification used | RRID:AB_398404 |
| Antibody | Anti-Occludin (Ms monoclonal) | LifeSpan Biosciences | Cat#: LS-B2320-50 | IF(1:2000)* *Tyramide signal amplification used | RRID:AB_1651895 |
| Antibody | Anti-Occludin (Rb monoclonal) | Abcam | Cat#: ab167161 | IHC (1:4000) | RRID:AB_2756463 |
| Antibody | Anti-Pericentrin (Ms monoclonal) | BD Biosciences | Cat#: 611814 | IHC (1:2000) | RRID:AB_399294 |
| Antibody | Anti-Gamma-Tubulin (Ms monoclonal) | Abcam | Cat#: ab27074 | IHC, IF (1:2000) | RRID:AB_2211240 |
| Antibody | Anti-Gamma-Tubulin (Rb polyclonal) | Abcam | Cat#: ab11317 | IHC, IF (1:2000) | RRID:AB_297921 |
| Antibody | Anti-Phospho-Beta-Catenin (Rb polyclonal) | Cell Signaling | Cat#: 9561 | WB (1:1000) | RRID:AB_331729 |
| Antibody | Anti-PhosphoH3 (Rb polyclonal) | EMD Millipore | Cat#: 06–570 | WB (1:2000) | RRID:AB_310177 |
| Antibody | Anti-PhosphoH3 (Ms monoclonal) | EMD Millipore | Cat#: 05–806 | IHC, IF (1:3000) | RRID:AB_310016 |
| Antibody | Anti-Transferrin (Ms monoclonal) | Thermo Fisher | Cat#: 13–6800 | WB (1:2000) | RRID:AB_86623 |
| Antibody | Anti-HDAC2 (Rb polyclonal) | Cell Signaling | Cat#: 2540 | WB (1:2000) | RRID:AB_2116822 |
| Antibody | Anti-GAPDH (Ms monoclonal) | Santa Cruz | Cat#: sc-365062 | WB (1:1000) | RRID:AB_10847862 |
| Antibody | Anti-PAX6 (Rb polyclonal) | Biolegend | Cat#: 901301 | IHC, IF (1:1000) | RRID:AB_2565003 |
| Antibody | Anti-TUJ1 (Ms monoclonal) | Biolegend | Cat#: 801201 | IHC, IF (1:3000) | RRID:AB_2313773 |
| Antibody | Anti-TBR2 (Rb polyclonal) | Abcam | Cat#: ab23345 | IHC (1:2000) | RRID:AB_778267 |
| Antibody | Anti-Ki67 (Ms monoclonal) | Thermo Fisher | Cat#: MA5-14520 | IHC (1:1000) | RRID:AB_10979488 |
| Antibody | Anti-CTIP2 (Rat monoclonal) | Abcam | Cat#: ab18465 | IHC (1:1000) | RRID:AB_2064130 |
| Antibody | Anti-CUX1 (Rb polyclonal) | Santa Cruz | Cat#: sc-13024 | IHC (1:1000) | RRID:AB_2261231 |
| Antibody | Anti-Activated Caspase-3 (Rb monoclonal) | Cell Signaling | Cat#: 9664 | IHC, IF (1:2000) | RRID:AB_2070042 |
| Antibody | Anti-ZO-1 (Ms monoclonal) | Thermo Fisher | Cat#: 33–9100 | IF (1:2000) | RRID:AB_2533147 |
| Antibody | Anti-anti-HOPX (Rb polyclonal) | Sigma | Cat#: HPA030180 | IF (1:2000) | RRID:AB_10603770 |
| Antibody | Anti-NeuroD1 (Ms monoclonal) | Abcam | Cat#: ab60704 | IF (1:2000) | RRID:AB_943491 |

Continued

| Reagent type (species) or resource | Designation | Source or reference | Identifiers | Additional information | RRID |
|---|---|---|---|---|---|
| Antibody | Anti-NuMA (Rb polyclonal) | Abcam | Cat#: ab84680 | IF (1:2000) | RRID:AB_2154610 |
| Antibody | Anti-Alpha-Tubulin (Rb monoclonal) | Abcam | Cat#: ab52866 | IF (1:3000) | RRID:AB_869989 |
| Antibody | Anti-Ran (Ms monoclonal) | BD Biosciences | Cat#: 610341 | WB: (1:2000) IF (1:4000)* *Tyramide signal amplification used | RRID:AB_397731 |
| Antibody | Anti-Acetylated Alpha Tubulin (Ms monoclonal) | Sigma | Cat#: T6793 | IF (1:4000) | RRID:AB_477585 |
| Antibody | Anti-PKC iota (aPKC) (Ms monoclonal) | BD Biosciences | Cat#: 610175 | IHC (1:1000) | RRID:AB_397574 |
| Antibody | Anti-PARD3 (Rb polyclonal) | EMD Millipore | Cat#: 07–330 | IHC (1:1000) | RRID:AB_2101325 |
| Antibody | Anti-Beta-Catenin (Rb polyclonal) | Cell Signaling | Cat#: 9562 | IHC (1:1000) | RRID:AB_331149 |
| Recombinant DNA reagent | pSpCas9(BB)—2A-Puro | Addgene | Plasmid #62988 | | RRID:Addgene_62988 |
| Commercial assay or kit | MycoAlert mycoplasma detection kit | Lonza | Cat#: LT07-318 | | |
| Commercial assay or kit | Tyramide Signal Amplification | Thermo Fisher | Cat#: B40956 | | |
| Commercial assay or kit | Subcellular Fractionation for Cells | Thermo Fisher | Cat#: 78840 | | |
| Commercial assay or kit | Subcellular Fractionation for Tissue | Thermo Fisher | Cat#: 87790 | | |
| Commercial assay or kit | iScript cDNA synthesis kit | BioRad USA | Cat#: 1708890 | | |
| Commercial assay or kit | Pierce BCA assay | Thermo Fisher | Cat#: 23225 | | |
| Other | Accutase | StemCell Technologies | Cat#: 07920 | 1X | |
| Other | Aggrewell plates | StemCell Technologies | Cat#: 34811 | | |
| Other | DMEM/F-12 Medium | Thermo Scientific | Cat#: 11320033 | | |
| Other | B-27 serum without Vitamin A | Thermo Scientific | Cat#: 12587010 | 1X | |
| Other | N2 supplement | Thermo Scientific | Cat#: 17502048 | 1X | |
| Other | GlutaMax | Thermo Scientific | Cat#: 35050061 | 1X | |
| Other | MEM Non-essential Amino Acid | Thermo Scientific | Cat#: 11140050 | 1X | |
| Other | penicillin-streptomycin | Thermo Scientific | Cat#: 15140122 | 100 U/ml | |
| Other | Neurobasal Medium | Thermo Scientific | Cat#: 21103049 | | |
| Other | B-27 supplement without Vitamin A | Thermo Scientific | Cat#: 12587010 | 1X | |
| Other | GlutaMax | Thermo Scientific | Cat#: 35050061 | 1X | |
| Other | 2-mercaptoethanol | Thermo Scientific | Cat#: 21985023 | 1:1000 | |
| Other | SB-431542 | StemCell Technologies | Cat#: 72234 | | |
| Other | LDN193189 | StemCell Technologies | Cat#: #72147 | | |

*Continued*

| Reagent type (species) or resource | Designation | Source or reference | Identifiers | Additional information | RRID |
|---|---|---|---|---|---|
| Other | ROCK inhibitor Y-27632 | StemCell Technologies | Cat#: 72304 | 10 µM | |
| Other | RIPA buffer | Thermo Fisher | Cat # 89900 | | |
| Peptide, recombinant protein | bFGF | Thermo Fisher | PHG0261 | 20 ng/mL | |
| Peptide, recombinant protein | EGF | Thermo Fisher | PHG0311 | 20 ng/mL | |
| Peptide, recombinant protein | BDNF | Peprotech | 450–02 | 20 ng/ml | |
| Peptide, recombinant protein | NT3 | Peprotech | 450–03 | 20 ng/ml | |
| Chemical compound | 1-bromo-3-chloropentane | Sigma | B62404 | | |

IHC=immunohistochemistry; IF=immunofluorescence; WB=Western Blot; Ms=mouse; Rb=rabbit.

### *Ocln* mutant mice

The *Ocln*$^{\Delta N/+}$ mouse line was generously provided by Dr. Margaret Neville (University of Colorado, Denver) and the colony was bred and maintained according to protocols approved by the Institutional Animal Care and Use Committee of Weill Cornell Medical College. Since homozygous mutants were reported to have difficulties breeding (*Saitou et al., 2000*), heterozygous mutants were bred to yield wild-type, heterozygous, and homozygous embryos.

### hESC culture

The human hESC line H9 (WISC-09) was maintained in feeder-free conditions on vitronectin-coated plates in mTESR1 maintenance media (StemCell Technologies, Vancouver, Canada). Cells were routinely passaged with ReLeSR (StemCell Technologies) according to manufacturer's instructions. Chromosome integrity of hESCs was confirmed by G-banding karyotype analyses performed in the pathology laboratory at Weill Cornell Medical College. Mycoplasma testing was carried out using a MycoAlert kit (Lonza). Briefly, cells at 70% confluency were washed once with 1x PBS, incubated with ReLESR for 1 min at 25°C, and once ReLESR was removed, the cells were incubated dry for 3 min at 37°C. Cells were then resuspended in warm mTESR1 media and transferred to freshly-coated plate at a dilution of 1:10 to 1:15.

### CRISPR mutagenesis

We employed CRISPR/Cas9 mutagenesis to target OCLN in hESC lines based on established protocol (*Ran et al., 2013*). Custom guide RNAs (sgRNAs) were designed using http://chopchop.cbu.uib.no/ and were cloned into pSpCas9(BB)−2A-Puro vector (pX459; Addgene). pSpCas9(BB)−2A-Puro vector (see Key Resource Table) was transfected in H9 hESCs plated at 50–70% confluency, using Amaxa Human Stem Cell Nucleofector kit (Lonza VPH-5002). Cells positive for the pX459 vector were selected for after 48 hr puromycin treatment (0.5 ug/mL) and the surviving cells were re-seeded in 96-well plates at clonal dilution and wells containing single clones were marked. After 7 days of culture, clonal colonies were harvested, gDNA was extracted, and PCR-amplified using primers found in to validate homozygous mutation (*Supplementary file 1*). Potential off-target effects were assessed using primers in *Supplementary file 1*.

### Cortical organoid differentiation

Cortical organoids (aka spheroids) were generated according to a previously published protocol (*Paşca et al., 2015*) with several adjustments. Adhered hESCs at 70–80% confluency were dissociated to single cells with 1X Accutase for 10 min at 37°C. Suspended single cells were transferred to Aggrewell plates at a density of $2.0 \times 10^6$ cells per well to ensure uniform starting size of spherical organoids. Cells were cultured in Forebrain Neural Induction Medium (FNIM) containing DMEM/F-

12, 1X B-27 serum without Vitamin A, 1X N2 supplement, 1X GlutaMax, 1X MEM Non-essential Amino Acid, 100 U/ml penicillin-streptomycin, and 2-mercaptoethanol. For neural induction, the media was supplemented with SB-431542 and LDN193189 for the first 5 days of culture (d0-d5). In addition, to promote single-cell survival, medium was the medium was supplemented with 10 μM ROCK inhibitor Y-27632 for the first 24 hr (d0). On d6, floating organoids were transferred to ultra-low attachment six-well plates in FNIM supplemented with 20 ng/mL bFGF and 20 ng/mL EGF, with daily media changes from d6-16 and media changes every other day from d17 to d25. On d25, media was replaced with Cortical Spheroid Media (CSM) containing Neurobasal (B-27 supplement without Vitamin A, 1X GlutaMax, 1X MEM Non-essential Amino Acid, 100 U/ml penicillin-streptomycin, and 2-mercaptoethanol, supplemented with 20 ng/ml BDNF and 20 ng/ml NT3 from d25 to d40, with media changes every other day.

## Immunohistochemistry

E10.5-E16.5 embryos were drop-fixed in 4% PFA in PBS at 4°C for 12–16 hr, cryoprotected in 15% then 30% sucrose for a total of 48 hr, then embedded in OCT and flash-frozen. Each block was cryosectioned at −20°C at 10–14 μm, depending on embryonic age, and was stored at −20°C until ready to be immunostained. P1, P7, and P20 pups were perfused with PBS followed by 4% PFA in PBS. Brains were harvested, drop-fixed in 4% PFA in PBS for 12 hr at 4°C, and cryoprotected as above.

Cortical organoids (d10, d20, or d40) were washed once with PBS, fixed in 4% PFA in PBS at 4°C for 12–16 hr, and cryoprotected in 30% sucrose for 48 hr. They were embedded in OCT in Tissue-Tek Cryomolds (10 × 10×5 mm) and flash-frozen (about 50 organoids were embedded per mold). Each frozen mold was cryosectioned at 10 μm and sections were stored at −20°C until immunostained.

Frozen sections were washed in PBS three times for 5 min each and were heated at 95°C for 15 min in antigen-retrieval buffer. They were then blocked for 1 hr with 10% Normal Donkey Serum (NDS) in PBS-Tween (0.1% tween) and incubated in primary antibody (see Key Resource Table) in blocking buffer at 4°C for 16–20 hr. Sections were washed with PBS (3 × 10 min) and incubated in secondary antibody in blocking buffer at room temperature for 1 hr. Sections were washed with PBS (3 × 10 min), incubated with DAPI for 10 min, and mounted using Prolong Antifade, based on manufacturer's instructions.

Immunocytochemistry hESCs were seeded on vitronectin-coated cover slips (Neuvitro #GG-18-pre) and were cultured until 50% confluent. Cells were washed once with PBS and fixed with 4% PFA in PBS for 20 min at room temperature. They were washed twice with PBS and heated at 95°C for 10 min in antigen-retrieval buffer. They were then blocked for 1 hr with 10% NDS in PBS-Tween (0.1% tween) and incubated in primary antibody (Key Resource Table) in blocking buffer at 4°C for 16–20 hr. Cover slips were washed with PBS (3 × 10 min) and incubated in secondary antibody in blocking buffer at room temperature for 1 hr. Cover slips were washed with PBS (3 × 10 min), incubated with DAPI for 10 min, and mounted using Prolong Antifade, based on manufacturer's instructions.

## Subcellular fractionation

Fractionation was performed using Thermo Fisher's kit for cultured cells and tissue following manufacturer's instructions (**Key Resource Table**). Adherent hESCs were manually harvested using cell scraper at 70% confluency, centrifuged at 500 x g for 5 min, and washed once with ice-cold PBS. The pellet was resuspended in ice-cold cytoplasmic extraction buffer (CEB) supplemented with 1x Protease inhibitor cocktail and centrifuged at 500 x g for 10 min. The supernatant (cytoplasmic extract) was collected into a clean tube, flash frozen and stored at −80°C. The remaining pellet was resuspended in ice-cold membrane extraction buffer (MEB) with 1x protease inhibitor, quickly vortexed, incubated at 4°C for 10 min, and centrifuged at 3000 x g for 5 min. The supernatant (membrane extract) was collected into clean tube, flash frozen, and stored at −80°C. The remaining pellet was resuspended in ice-cold nuclear extraction buffer (NEB) with 1x protease inhibitor, vortexed for 15 s, incubated at 4°C for 30 min, and centrifuged at 5000 x g for 5 min. The supernatant (soluble nuclear extract) was collected into clean tube, flash frozen, and stored at −80°C. The remaining pellet was resuspended in room-temperature NEB with 5 mM CaCl$_2$, three units/uL of micrococcal nuclease, and 1x protease inhibitor, vortexed for 15 s, incubated at 25°C for 15 min, and centrifuged at 16,000 x g for 5 min. The supernatant (chromatin-bound extract) was collected into clean tube,

flash frozen, and stored at −80℃. Fractionation of mouse tissues were performed as described above, but were first homogenized using a Polytron handheld homogenizer.

## RT-PCR

RNA was isolated from 3-D spheroids or from E12.5 mouse embryos using TRI reagent (Sigma T9424) per manufacturer's instructions. Briefly, cells or mouse tissue was homogenized, resuspended in 1 mL TRI reagent, and incubated at 25℃ for 5 min. 100 uL of 1-bromo-3-chloropentane was added for phase separation and after rigorous mixing and 10 min incubation at room temperature, the sample was centrifuged at 12,000 x g for 15 min at 4℃. The upper aqueous layer was transferred to a fresh tube and mixed with 500 μL of 100% isopropanol and centrifuged at 12,000 x g for 10 min at 4℃. Supernatant was discarded and the remaining pellet was resuspended in 75% ethanol in DEPC-treated water, centrifuged at 7500 x g for 5 min at 4℃. The RNA pellet was air-dried for 10–20 min, resuspended in DEPC-treated water, and flash-frozen for long-term storage at −80℃. Purified RNA was reverse transcribed using the iScript cDNA kit and PCR was performed using primers in *Supplementary file 1*.

## Western blotting

Mouse tissue or hESC were lysed in ice-cold RIPA buffer supplemented with 1x Protease and Phosphatase inhibitors. The concentration of the protein lysate was determined using BCA Protein Assay Reagent. 30–40 micrograms of protein were separated on a 4–12% NuPage Bis-Tris protein gels and transferred to 0.2 μm nitrocellulose membranes (Biorad). The membranes were blocked in Odyssey Blocking Buffer at 25℃ for 1 hr, incubated with primary antibodies at 4℃ overnight, washed three times with TBS, incubated with secondary antibodies for 1 hr, and imaged using Odyssey Imager.

## Acknowledgements

This work was supported by the National Institutes of Health: P01HD067244, R01NS105477 and the TriSci Stem Cell Initiative (MER) and the National Institutes of Health training grant: T32HD060600 (RMB).

## Additional information

### Funding

| Funder | Grant reference number | Author |
| --- | --- | --- |
| National Institutes of Health | P01HD067244 | M Elizabeth Ross |
| National Institutes of Health | R01NS105477 | M Elizabeth Ross |
| National Institutes of Health | Training Fellowship (T32HD060600) | Raphael M Bendriem |

The funders had no role in study design, data collection and interpretation, or the decision to submit the work for publication.

### Author contributions

Raphael M Bendriem, Conceptualization, Data curation, Formal analysis, Investigation, Visualization, Writing—original draft; Shawn Singh, Data curation, Validation, Writing—review and editing, Data acquisition; Alice Abdel Aleem, Conceptualization, Writing—review and editing; David A Antonetti, Conceptualization, Resources, Writing—review and editing; M Elizabeth Ross, Conceptualization, Resources, Data curation, Supervision, Funding acquisition, Investigation, Methodology, Project administration, Writing—review and editing

### Author ORCIDs

M Elizabeth Ross https://orcid.org/0000-0001-6440-8089

## Ethics

Animal experimentation: This study was performed in strict accordance with the recommendations in the Guide for the Care and Use of Laboratory Animals of the National Institutes of Health. All of the animals were handled according to an approved Weill Cornell Medical College institutional animal care and use committee (IACUC) protocol (#0711-683A). All tissue collection for histology was performed under anesthesia and every effort was made to minimize suffering.

## Decision letter and Author response

Decision letter https://doi.org/10.7554/eLife.49376.sa1
Author response https://doi.org/10.7554/eLife.49376.sa2

# Additional files

## Supplementary files

• Supplementary file 1. Primer sequence table.A compiled list of all primer sequences used in this study, cross-checked by experiment and figure number.

• Transparent reporting form

## Data availability

All data generated or analysed during this study are included in the manuscript and supporting files.

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
