## [Decision Letter]

**Acceptance summary:**

This study convincingly characterizes a previously unrecognized function for the tight junction protein Occludin (OCLN) at centrosomes/mitotic spindles to regulate the neuroepithelial to radial glial cell transition in vivo in the mouse and in human ESC-derived cortical organoids. Previous work reported localization of OCLN to centrosomes and regulation of mitosis. A seminal finding of this study is the demonstration of the expression characteristics of full length (FL) OCLN and a truncated form lacking the portion encoded by exons 2 and 3. It is the loss of OCLN-FL that impairs cortical development. Presumably via its interaction with the mitotic spindle protein NuMa the investigators nicely show that OCLN functions to stabilize mitotic spindles and thereby impacts M-phase length, apoptosis, and importantly the proper balance between neural progenitor pool size and differentiated neurons.

**Decision letter after peer review:**

Thank you for submitting your article "Tight junction protein occludin regulates progenitor self-renewal and survival in developing cortex" for consideration by *eLife*. Your article has been reviewed by three peer reviewers, one of whom is a member of our Board of Reviewing Editors, and the evaluation has been overseen by Marianne Bronner as the Senior Editor. The following individuals involved in review of your submission have agreed to reveal their identity: Sarah Lutz (Reviewer #2), Leslie M Thompson (Reviewer #3).

The reviewers discussed the reviews with one another and the Reviewing Editor has drafted this decision to help you prepare a revised submission.

Summary:

The results reported in this manuscript provide novel fundamental biological insight regarding the function of the protein OCLN that will be of broad interest for a readership focused on neurodevelopment. Mutations in OCLN, previously believed to be exclusively a tight-junction protein, result in microcephaly and cortical malformation. This study convincingly characterizes a previously unrecognized function for the tight junction protein Occludin (OCLN) at centrosomes/mitotic spindles to regulate the neuroepithelial to radial glial cell transition in vivo in the mouse and in human ESC-derived cortical organoids.

Essential revisions:

In Figure 2F – legend states that T-test was done. It is suggested that authors use ANOVA with posthoc for statistical analysis of multiple and timepoints groups.

Minor points:

1) Figure 3F, the X axis should go from 0-100 and Figure 1A "enlarged" lacks a scale bar.

2) The Discussion could be more focused on the major contribution of this study, i.e. loss of OCLN-FL that impairs cortical development. In this light, what is known about the human OCLN mutations that cause microcephaly – are they consistent with the importance of OCLN-FL in cortical development? In addition, less space in the Discussion should be devoted to aspects of OCLN in tight junctions and role of truncated OCLN, points that are not addressed by data in this manuscript.

3) The authors interpret their data to suggest that the phenotypes they observe are due to loss-of-function caused by loss of full length OCLN. Could an alternative explanation be that some or all of those phenotypes are caused by or contributed to by gain-of-function of OCLN-ΔN?

4) The authors interpret previous results showing that "*Ocln* mutant mice did not exhibit gross phenotypes that would be expected of impaired TJs" to mean that "OCLN plays no part in TJ maintenance or formation". Could an alternative explanation be that OCLN-ΔN is insufficient for TJ formation and maintenance? In general, it would be helpful for the authors to comment specifically about truncated forms of OCLN in both mouse and human and their roles. Especially since it is shown to be mainly nuclear localized and specifically localized at centrosomes. The authors make a strong case for the full-length protein but a comment in the Discussion about the truncated version and its roles would be interesting to readers particularly if there are thoughts on whether full length is playing similar role at centrosome as truncated.

5) Some OCLN staining presented throughout the manuscript was interpreted to be localization to the plasma membrane, but appears to also localize to the cytosol. For instance, in Figure 2C there appears to be some diffuse nuclear staining of n-term OCLN in the ΔN/ΔN mice strains. This is visible in all cells from 10.5 to 12.5. Additionally, OCLN is a well-known tight junction protein, but does not appear to show such localization – is this a temporal effect during development?

6) Some figure legends are missing information such as how many mice per genotype were assessed per figure. What do the red arrows show in Figure 3E? There doesn't appear to be DAPI staining in Figure 4A, but it is listed in the legend. In Figure 6B, please indicate how many spheroids were analyzed, how many experiments were performed, and what the error bars represent. In Figure 6G, what do the error bars represent? In Figure 7, please indicate antibody as it was shown to be important in previous figures.

7) In the Abstract there is little mention of temporal expression differences of OCLN isoforms. This is an interesting finding from this manuscript that isn't highlighted until the final sentence.

8) Gene and protein names don't always appear to be written appropriately for the species being presented/discussed. A few examples include OCLN when presenting mouse data.

9) Figure 1H should have the axis range from 0-1.45. Rationale is that the magnitude of the reduction in cortical size throughout the paper is small – which is fine, it is small but carefully characterized and phenotypically important.

10) Please provide double immunostaining of full length and truncated OCLN and a vascular marker. This could be done with existing mouse brain sections such as those used in figures looking at a section of cortex, etc.

[Editors' note: further revisions were requested prior to acceptance, as described below.]

Congratulations, we are pleased to inform you that your article, "Tight junction protein occludin regulates progenitor self-renewal and survival in developing cortex", has been provisionally accepted for publication in *eLife*.

The manuscript has been improved but there are some remaining issues that need to be addressed before publication, as outlined below:

1) For the following response to review:

In the authors’ response to point 5: “While it is known that OCLN is transported to and from various subcellular locations as part of normal intracellular cell processing, reports-including IP-mass spectrometry proteomic screens (Fredriksson et al., 2014) have failed to detect an intranuclear function of OCLN (i.e. transcription factor, nuclear protein interaction, etc.). This diffuse signal the reviewer refers to is likely background, non-specific antibody labeling accentuated at higher magnifications that focus on a single cell. Indeed, in the lower magnification images in Figure 1, the nuclei appear as voids in the OCLN antibody stains.”

Upon cursory examination of Fredriksson et al., 2015 (assuming the correct publication was found), IP-MS does not appear to have been performed on the mouse model used here (ΔN/ΔN OCLN), preventing correlation of the findings in their report with those of this manuscript. The authors imply that the nuclear signal observed is accentuated at the higher magnification, however clear nuclear signal is present even at lower magnification, specifically in ΔN/ΔN animals at E10.5 (Figure 2C top left panel). Without proper controls (secondary antibody alone), the response that the nuclear signal (specifically in Figure 2C) is "likely background" cannot actually be verified. Additionally, the statement that nuclei void of OCLN staining in Figure 1 supports these claims cannot be verified – Figure 1 states that mAb and pAb against OCLN were used while Figure 2C states that an N-term OCLN antibody was used. With the information given, reviewers are unable to verify whether the same antibody was used in these figures. While additional staining would clarify, we would only request that a revision of the text to support perhaps unknown and undescribed nuclear function for OCLN in ΔN/ΔN animals.

2) In the authors’ response to point 8: “Care has been taken to ensure that OCLN is appropriately referring to encoded protein in all instances in the manuscript.”

Some figures need edits (e.g. Figure 1) to capitalize proteins, for consistency.

One last thing – in their list of figure legends they listed "Source Data" for certain figures. They took out a lot of the experimental details (numbers of mice etc.) and indicated in the rebuttal that it was included. We are assuming it is in source data, but we did not see those files.

---

## [Author Response]

Essential revisions:In Figure 2F – legend states that T-test was done. It is suggested that authors use ANOVA with posthoc for statistical analysis of multiple and timepoints groups.

The authors have changed the statistical analysis performed for Figure 2F to two-way ANOVA with Sidak’s multiple comparison test, comparing WT and mutant within each age group.

Minor points:1) Figure 3F, the X axis should go from 0-100 and Figure 1A "enlarged" lacks a scale bar.

The x-axis for Figure 3F has been updated to match reviewers’ comments. A scale bar has been added to the “enlarged” image in Figure 1A; it represents 25 microns, which is the same length represented by all other scale bars in Figure 1A.

2) The Discussion could be more focused on the major contribution of this study, i.e. loss of OCLN-FL that impairs cortical development. In this light, what is known about the human OCLN mutations that cause microcephaly – are they consistent with the importance of OCLN-FL in cortical development? In addition, less space in the Discussion should be devoted to aspects of OCLN in tight junctions and role of truncated OCLN, points that are not addressed by data in this manuscript.

The authors agree that a more extensive discussion of the clinical importance of OCLN-FL is necessary – we have updated the discussion accordingly and have shortened the discussion regarding OCLN in the context of tight junctions.

3) The authors interpret their data to suggest that the phenotypes they observe are due to loss-of-function caused by loss of full length OCLN. Could an alternative explanation be that some or all of those phenotypes are caused by or contributed to by gain-of-function of OCLN-ΔN?

While the observed phenotypes could be caused, in part, by a dominant negative effect of the truncated protein (OCLN-ΔN) rather than loss-of-function of full-length OCLN, this is unlikely for several reasons. First, this human disorder is associated in half of reported cases with autosomal recessively inherited, truncating mutations in exon 3, which speaks against a dominant negative effect of the truncated isoform, since heterozygous individuals are asymptomatic. Second, a number of reported insertion/deletion *OCLN* mutations causing BLC-PMG were found on exons 5, 6, or 7 that would result in loss-of-function of all OCLN isoforms. Third, there is no indication yet to suggest a genotype-phenotype correlation with clinical severity between the more proximal vs. distal *OCLN* mutations. Together these features strongly indicate that loss-of-function of full-length OCLN underlies the phenotypes observed in this study.

The Discussion section has been updated to clarify this point.

4) The authors interpret previous results showing that "Ocln mutant mice did not exhibit gross phenotypes that would be expected of impaired TJs" to mean that "OCLN plays no part in TJ maintenance or formation". Could an alternative explanation be that OCLN-ΔN is insufficient for TJ formation and maintenance? In general, it would be helpful for the authors to comment specifically about truncated forms of OCLN in both mouse and human and their roles. Especially since it is shown to be mainly nuclear localized and specifically localized at centrosomes. The authors make a strong case for the full-length protein but a comment in the Discussion about the truncated version and its roles would be interesting to readers particularly if there are thoughts on whether full length is playing similar role at centrosome as truncated.

The authors did not intend to imply that OCLN plays no part in TJ maintenance or formation. In light of the observations that the OCLN-ΔN is continually expressed in the *Ocln* mouse mutant, a closer look at the role that this truncated isoform may play in TJ formation is warranted. The Discussion has been updated to clarify this point.

Very little is known regarding the function of OCLN-ΔN and other shorter isoforms. The data and results analyzed in this study revolve around the role of full-length OCLN. Based on recent reports of astral microtubule function and the astral microtubule disruption demonstrated here in cells lacking full-length OCLN, the authors suggest that this isoform is required for proper mitotic centrosomal positioning to ensure a near perpendicular cleavage plane of symmetrically dividing radial glial cells. The authors thought it best to minimize the amount of text dedicated to the presumed roles of truncated isoforms in the absence of data beyond OCLN-interaction with pericentrosomal matrix proteins NuMA and RAN. However, we fully support any future endeavors to tease out the role(s) of these truncated isoforms in neurodevelopment and neurovascular biology.

5) Some OCLN staining presented throughout the manuscript was interpreted to be localization to the plasma membrane, but appears to also localize to the cytosol. For instance, in Figure 2C there appears to be some diffuse nuclear staining of n-term OCLN in the ΔN/ΔN mice stains. This is visible in all cells from 10.5 to 12.5. Additionally, OCLN is a well-known tight junction protein, but does not appear to show such localization – is this a temporal effect during development?

While it is known that OCLN is transported to and from various subcellular locations as part of normal intracellular cell processing, reports – including IP-mass spectrometry proteomic screens (Fredriksson et al.,2014) have failed to detect an intranuclear function of OCLN (i.e. transcription factor, nuclear protein interaction, etc.). This diffuse signal the reviewer refers to is likely background, non-specific antibody labeling accentuated at higher magnifications that focus on a single cell. Indeed, in the lower magnification images in Figure 1, the nuclei appear as voids in the OCLN antibody stains.

6) Some figure legends are missing information such as how many mice per genotype were assessed per figure. What do the red arrows show in Figure 3E? There doesn't appear to be DAPI staining in Figure 4A, but it is listed in the legend. In Figure 6B, please indicate how many spheroids were analyzed, how many experiments were performed, and what the error bars represent. In Figure 6G, what do the error bars represent? In Figure 7, please indicate antibody as it was shown to be important in previous figures.

The authors have included the missing information requested by reviewers. Specifically, we have specified how many mice were analyzed in Figure 4D, we have specified what the red arrows signify in Figure 3E, we have fixed the figure legend for Figures 4A and 4C taking into account lack of DAPI presence, we have specified the number of organoids analyzed per group in Figure 6B, and we have specified what error bars represent in Figures 6B and 6G. All antibodies used can be found in the Key Resource Table.

7) In the Abstract there is little mention of temporal expression differences of OCLN isoforms. This is an interesting finding from this manuscript that isn't highlighted until the final sentence.

The authors agree that this is an interesting and important aspect of OCLN isoform developmental regulation. It was not emphasized in the Abstract due to word limitations. The Abstract has been revised to highlight this feature while streamlining the text to maintain the 150 word limit.

8) Gene and protein names don't always appear to be written appropriately for the species being presented/discussed. A few examples include OCLN when presenting mouse data.

The accepted convention for nomenclature is to italicize in all capital letters to indicate human genes while only the first letter is capitalized for mouse genes. However, the encoded proteins should be in all caps, no italics, for human and mouse proteins. [See https://www.biosciencewriters.com/Guidelines-for-Formatting-Gene-and-Protein-Names.aspx]. Care has been taken to ensure that OCLN is appropriately referring to encoded protein in all instances in the manuscript.

9) Figure 1H should have the axis range from 0-1.45. Rationale is that the magnitude of the reduction in cortical size throughout the paper is small – which is fine, it is small but carefully characterized and phenotypically important.

The reviewer refers to Figure 2H. The panel has been revised so that the axis range extends from 0-1.45.

10) Provide double immunostaining of full length and truncated OCLN and a vascular marker. Could be done with existing mouse brain sections such as those used in figures looking at a section of cortex, etc.

Immunolabeling with C-terminal-specific OCLN antibody (recognizing both full-length and truncated isoforms of OCLN) and co-labeled with vascular endothelial antigen, CD31, antibody is provided in Figure 2—figure supplement 1. Sections from *Ocln^+/+^* and *Ocln^∆N/∆N^* siblings at E12.5 and E14.5 are shown. Panels show robust labeling of full length OCLN in vessels of wildtype embryos at E12.5 and E14.5, but not in vessels of the mutant animals, though OCLN labeling of mitotic spindles of dividing progenitors is seen in both genotypes.

[Editors' note: further revisions were requested prior to acceptance, as described below.]

The manuscript has been improved but there are some remaining issues that need to be addressed before publication, as outlined below:1) For the following response to review:In the authors’ response to point 5: “While it is known that OCLN is transported to and from various subcellular locations as part of normal intracellular cell processing, reports-including IP-mass spectrometry proteomic screens (Fredriksson et al., 2014) have failed to detect an intranuclear function of OCLN (i.e. transcription factor, nuclear protein interaction, etc.). This diffuse signal the reviewer refers to is likely background, non-specific antibody labeling accentuated at higher magnifications that focus on a single cell.Indeed, in the lower magnification images in Figure 1, the nuclei appear as voids in the OCLN antibody stains.”Upon cursory examination of Fredriksson et al., 2015 (assuming the correct publication was found), IP-MS does not appear to have been performed on the mouse model used here (ΔN/ΔN OCLN), preventing correlation of the findings in their report with those of this manuscript. The authors imply that the nuclear signal observed is accentuated at the higher magnification, however clear nuclear signal is present even at lower magnification, specifically in ΔN/ΔN animals at E10.5 (Figure 2C top left panel). Without proper controls (secondary antibody alone), the response that the nuclear signal (specifically in Figure 2C) is "likely background" cannot actually be verified. Additionally, the statement that nuclei void of OCLN staining in Figure 1 supports these claims cannot be verified – Figure 1 states that mAb and pAb against OCLN were used while Figure 2C states that an N-term OCLN antibody was used. With the information given, reviewers are unable to verify whether the same antibody was used in these figures. While additional staining would clarify, we would only request that a revision of the text to support perhaps unknown and undescribed nuclear function for OCLN in ΔN/ΔN animals.

The reviewer makes an excellent point, and we have accordingly modified the Results section to comment that diffuse OCLN nuclear labeling was detected in VZ cells. We add in the Discussion that we cannot with our present data rule out the possibility of a nuclear role for one or more OCLN isoform and note that OCLN has been found in a previously reported proteomic screen to associate with a nuclear envelope protein, ezrin.

2) In the authors’ response to point 8: Care has been taken to ensure that OCLN is appropriately referring to encoded protein in all instances in the manuscript.Some figures need edits (e.g. Figure 1) to capitalize proteins, for consistency.

We have modified the figures to ensure that labeling of the panels consistently use all capitals for protein abbreviations

One last thing – in their list of figure legends they listed "Source Data" for certain figures. They took out a lot of the experimental details (numbers of mice etc) and indicated in the rebuttal that it was included. We are assuming it is in source data, but we did not see those files.

Indeed, this information [numbers of mice, biological replicates, etc.] is included in the source data. We have added this information to the figure legends for ready access.

We thank the reviewers for their many helpful comments that have significantly improved the manuscript. We hope that it is now suitable for final approval and publication in *eLife*.